# MOBO-OSD: Batch Multi-Objective Bayesian Optimization via Orthogonal Search Directions

**Lam Ngo**
School of Computing Technologies
RMIT University, Australia
`tung.lam.ngo@rmit.edu.au`

**Huong Ha**
School of Computing Technologies
RMIT University, Australia
`huong.ha@rmit.edu.au`

**Jeffrey Chan**
School of Computing Technologies
RMIT University, Australia
`jeffrey.chan@rmit.edu.au`

**Hongyu Zhang**
School of Big Data and Software Engineering
Chongqing University, China
`hyzhang@cqu.edu.cn`

## Abstract

Bayesian Optimization (BO) is a powerful tool for optimizing expensive black-box objective functions. While extensive research has been conducted on the single-objective optimization problem, the multi-objective optimization problem remains challenging. In this paper, we propose MOBO-OSD, a multi-objective Bayesian Optimization algorithm designed to generate a diverse set of Pareto optimal solutions by solving multiple constrained optimization problems, referred to as *MOBO-OSD subproblems*, along orthogonal search directions (OSDs) defined with respect to an approximated convex hull of individual objective minima. By employing a well-distributed set of OSDs, MOBO-OSD ensures broad coverage of the objective space, enhancing both solution diversity and hypervolume performance. To further improve the density of the set of Pareto optimal candidate solutions without requiring an excessive number of subproblems, we leverage a Pareto Front Estimation technique to generate additional solutions in the neighborhood of existing solutions. Additionally, MOBO-OSD supports batch optimization, enabling parallel function evaluations to accelerate the optimization process when resources are available. Through extensive experiments and analysis on a variety of synthetic and real-world benchmark functions with two to six objectives, we demonstrate that MOBO-OSD consistently outperforms the state-of-the-art algorithms. Our code implementation can be found at `https://github.com/LamNgo1/mobo-osd`.

## 1 Introduction

Multi-objective Bayesian Optimization (MOBO) has recently attracted much attention [5, 13, 52, 44, 1]. MOBO extends Bayesian Optimization (BO) — a powerful framework for optimizing expensive black-box functions — to settings where multiple, often conflicting, objectives must be optimized simultaneously. Applications of MOBO include but are not limited to machine learning [47, 29], material design [18, 56], agriculture [27, 28], robotics [54, 34] and vehicle design [32, 2].

There are several challenges when working with MOBO problems. First, the objective functions are often conflicting, meaning that improving one objective may deteriorate another. Hence the goal is not to identify a single optimal solution, but rather a set of *Pareto optimal solutions*. Second, maintaining the diversity among these Pareto optimal solutions is crucial to capture the wide range of trade-off across all objectives. This becomes more challenging under the limited evaluation budget in MOBO settings, where there is less budget to balance between exploration and exploitation while

39th Conference on Neural Information Processing Systems (NeurIPS 2025).

covering the Pareto front. Third, batch optimization plays an important role in reducing the time and cost through parallel evaluations of objective functions. However, in MOBO, it introduces additional complexity, as it requires modeling the interactions among unobserved points across multiple objective functions. Fourth, the computational cost remains a significant drawback, as the cost for training surrogate models and computing multi-objective acquisition functions, e.g., EHVI [13], become increasingly expensive with the number of objectives. Despite the growing body of MOBO research, there exists substantial room for improvement. Many existing approaches rely on scalarization techniques [5, 9, 41], which often fail to capture a diverse Pareto front, leading to suboptimal performance in terms of the hypervolume indicator. Moreover, several studies are unable to scale to a large number of objectives ($M > 3$) [33] or suffer from high computational cost in batch optimization settings [13, 52].

In this paper, our goal is to develop a novel MOBO algorithm that prioritizes Pareto front diversity, resulting in an improvement in hypervolume performance and scalability to an arbitrary number of objectives in both sequential and batch optimization settings. To achieve this, we build upon the key insight of the Normal Boundary Intersection (NBI) method [11]: the intersection points between the boundary of the objective space and the vectors orthogonal to the *convex hull of individual minima* (CHIM) of the objectives could be Pareto optimal solutions. This geometric approach has been shown to generate a well-distributed (*diverse*) set of Pareto optimal solutions and scale well to a large number of objectives in multi-objective optimization problems with a large evaluation budget. To incorporate this idea under the limited evaluation budget in MOBO problems, we first introduce a technique to effectively approximate the CHIM. We then generate a well-distributed set of *orthogonal search directions* (OSDs) with respect to the approximated CHIM and define a tailored search procedure that solves a constrained single-objective optimization problem along each OSD (*MOBO-OSD subproblem*) to identify intersection points with the boundary that are likely to be Pareto optimal. By constructing a well-distributed set of OSDs, MOBO-OSD achieves broad coverage of the objective space, enhancing both solution diversity and hypervolume performance. To further enrich the set of Pareto optimal candidate solutions without the need to solve a large number of MOBO-OSD subproblems, we incorporate a Pareto Front Estimation (PFE) technique [46] that locally explores the neighborhood of existing Pareto optimal candidates. All candidates are aggregated and evaluated using the hypervolume improvement acquisition function [19], and those with the highest scores are selected as the next batch of data points for evaluation. Finally, to support batch optimization, we incorporate the Kriging Believer technique [22] while considering the diversity of exploration regions, i.e., ensuring that selected points for observation originate from different exploration spaces. We name our method *Batch Multi-Objective Bayesian Optimization via Orthogonal Search Directions* (MOBO-OSD). Our thorough analysis and extensive experiments on various synthetic and real-world benchmark problems, in both sequential and batch optimization settings, show that MOBO-OSD outperforms the state-of-the-art MOBO methods. We summarize our contributions as follows.

- We propose MOBO-OSD, a novel MOBO algorithm that generates a diverse set of Pareto optimal candidate solutions by solving multiple optimization subproblems defined along search directions orthogonal to the approximated CHIM of the objective space. To further enhance the diversity, we propose to locally explore the Pareto set for additional Pareto solutions via a PFE technique.
- We develop MOBO-OSD in such a way that it can perform effectively in the batch optimization setting by leveraging the Kriging Believer technique and exploration space information.
- We demonstrate that MOBO-OSD can outperform the state-of-the-art MOBO baselines on a comprehensive set of synthetic and real-world multi-objective benchmark problems with a wide range of numbers of objectives, in both sequential and batch settings.

## 2 Background

### 2.1 Multi-Objective Optimization

We consider a multi-objective optimization (MOO) problem involving a *vector-valued* objective function $\mathbf{f} : \mathcal{X} \to \mathcal{Y}$ with $\mathbf{f} = (f_1, \ldots, f_M)$, where $\mathcal{X} \subset \mathbb{R}^D$ is a $D$-dimensional input space, and $\mathcal{Y} \subset \mathbb{R}^M$ is an $M$-dimensional output space ($M > 1$). Without loss of generality, we assume the goal is to minimize all objectives of $\mathbf{f}$. In MOO, it is generally not possible to find a single solution that optimizes all objectives simultaneously. Instead, the aim is to identify the set of *Pareto optimal solutions*, where no objective can be improved without deteriorating at least one of the others.

A solution $\mathbf{f}(\mathbf{x}^{(i)})$ is said to *Pareto dominate* another solution $\mathbf{f}(\mathbf{x}^{(j)})$, i.e., $\mathbf{f}(\mathbf{x}^{(i)}) \succ \mathbf{f}(\mathbf{x}^{(j)})$, if $f_m(\mathbf{x}^{(i)}) \geq f_m(\mathbf{x}^{(j)}) \ \forall m = 1, \ldots, M$ and there exists $m' \in \{1, \ldots, M\}$ such that $f_m(\mathbf{x}^{(i)}) > f_m(\mathbf{x}^{(j)})$. The set of Pareto optimal solutions is called the *Pareto front* $\mathcal{P}_f = \{\mathbf{f}(\mathbf{x}) \mid \nexists \mathbf{x}' \in \mathcal{X} : \mathbf{f}(\mathbf{x}') \succ \mathbf{f}(\mathbf{x})\}$, and the corresponding set of Pareto optimal inputs is called the *Pareto set* $\mathcal{P}_s = \{\mathbf{x} \in \mathcal{X} \mid \mathbf{f}(\mathbf{x}) \in \mathcal{P}_f\}$. Formally, the MOO problem is expressed as finding the Pareto front $\mathcal{P}_f$ and Pareto set $\mathcal{P}_s$ such that,

$$\mathcal{P}_f \in \min_{\mathbf{x} \in \mathcal{X}} (f_1(\mathbf{x}), f_2(\mathbf{x}), \ldots, f_M(\mathbf{x})). \tag{1}$$

To measure the quality of a Pareto front $\mathcal{P}_f$, the *Hypervolume* (HV) indicator [59] is one of the most widely used metrics [45, 5, 9, 13, 4, 1]. Given a reference point $\mathbf{r} \in \mathbb{R}^M$, the HV indicator of a finite approximated Pareto front $\mathcal{P}_f$ is defined as the $M$-dimensional Lebesgue measure $\lambda_M$ of the space dominated by solutions $\mathbf{p}$ in $\mathcal{P}_f$ and upper bounded by the reference point $\mathbf{r}$. Formally, $\mathrm{HV}(\mathcal{P}_f, \mathbf{r}) = \lambda_M(\bigcup_{\mathbf{p} \in \mathcal{P}_f}[\mathbf{r}, \mathbf{p}])$, where $[\mathbf{r}, \mathbf{p}]$ denotes the hyperrectangle bounded by the reference point $\mathbf{r}$ and $\mathbf{p} \in \mathcal{P}_f$ [9, 13, 14]. The higher the HV, the better $\mathcal{P}_f$ approximates the true Pareto front.

## 2.2 Multi-Objective Bayesian Optimization

Bayesian Optimization [21] is a common tool for optimizing *expensive black-box* objective functions $f$. Given a minimization problem, the goal is to find the global optimum of the function $f$ using the least function evaluations. BO sequentially selects observation data via an iterative process. Each BO iteration trains a probabilistic *surrogate model*, builds an *acquisition function*, then selects the acquisition function's maximizer as the next observation. The most common type of surrogate model for BO is a Gaussian Process (GP) [55], which provides a posterior distribution over the objective function given the observed dataset $\mathcal{D}$. The acquisition function $\alpha : \mathcal{X} \to \mathbb{R}$ is constructed from the surrogate model and an optimization policy to quantify the utility of each unobserved data point. Single-objective BO has many common acquisition functions such as EI [37], UCB [48] and TS [51].

Multi-objective Bayesian Optimization (MOBO) extends the capabilities of BO to optimize *expensive black-box vector-valued* objective functions $\mathbf{f} : \mathcal{X} \to \mathcal{Y}$ with $\mathbf{f} = (f_1, \ldots, f_M)$. Given a minimization problem, the goal is to find the Pareto set $\mathcal{P}_s$ and the corresponding Pareto front $\mathcal{P}_f$, *using the least function evaluations*. In MOBO, the most common surrogate model is a set of GPs, where each GP independently represents an objective function $f_m$ [31, 9, 41, 5, 13]. For acquisition functions, while several works leverage existing single-objective acquisition functions [31, 41, 5], others propose new acquisition functions tailored for the MOO setting [23, 4, 52, 13, 14, 12].

## 3 Related Work

There have been various works aiming to develop MOBO algorithms. Many works have attempted to adapt single-objective (SO) acquisition functions to the multi-objective optimization (MOO) framework. One of the earliest methods is ParEGO [31], which randomly scalarizes the objectives and applies the EI acquisition function [37] to determine the next data points for observation. Paria et al. [41] systematically generalize the random scalarization technique, allowing different scalarization techniques, e.g., weighted sum and Tchebyshev [39], as well as different SO acquisition functions, e.g., TS [51], UCB [48], to be integrated into the MOBO framework. TSEMO [9] formulates a multi-objective TS acquisition function [51] by sampling all objectives, employs NSGA-II [16] to solve the resulting optimization problem, and selects the next evaluations by maximizing hypervolume improvement. USeMO [5] aims to maximize the uncertainty reduction of the candidate points generated by a MO acquisition function, which is defined by applying SO acquisition functions to each objective GP. Recently, PDBO [1] uses a multi-armed bandit technique to select an SO acquisition function and generate a candidate pool, followed by a Determinantal Point Process to select a diverse set of solutions. Our proposed method operates directly on a multi-objective (MO) acquisition function, the Hypervolume Improvement, which is more suitable for the MOBO setting. In the experimental section, we compare against ParEGO, USeMO and PDBO - one of the latest state-of-the-art methods.

There are also many works that have focused on developing new acquisition functions tailored for the MOO setting. Information-theoretic (IT) MO acquisition functions aim to maximize the information gain of the next observation about the objective functions. Examples include PESMO [23],

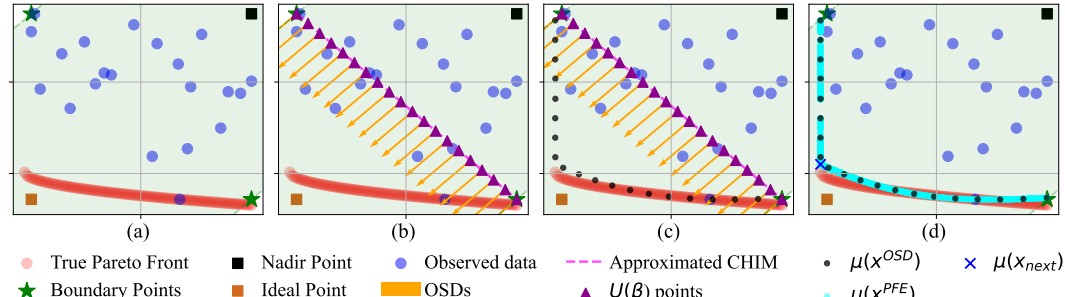

Figure 1: **Summary of the MOBO-OSD algorithm**: **(a)** The boundary points are defined via the nadir and ideal points; **(b)** The approximated CHIM is defined from the boundary points and the OSDs are defined on each $\mathcal{U}(\boldsymbol{\beta})$ point and orthogonal to the approximated CHIM; **(c)** The MOBO-OSD subproblem (Eq. (2)) is optimized for $\mathbf{x}^{\text{OSD}}$ candidates; **(d)** Additional candidates $\mathbf{x}^{\text{PFE}}$ are generated by locally exploring around $\mathbf{x}^{\text{OSD}}$. Finally, $\mathbf{x}_{\text{next}}$ is chosen based on Eq. (4).

MESMO [4], PFES [49], and JES [52]. PAL [61] is another algorithm based on information theory, but it is only applicable to input spaces with finite set of discrete points. Another widely used approach to define MO acquisition functions is through the HV indicator. The EHVI acquisition function [19] extends the EI concept from SOO to compute the expected improvement in the hypervolume of the Pareto front. Daulton et al. [13] propose a novel formulation of EHVI in a parallel setting, namely qEHVI, based on Monte-Carlo sampling. Despite being differentiable, even for $M > 2$ objectives, qEHVI can suffer from high computational cost. Unlike qEHVI, which computes the expected hypervolume improvement under the posterior distribution, DGEMO [33] relies solely on the hypervolume improvement of the posterior mean, yet its integration with the Pareto Front Estimation technique [46] renders the method competitive. Nonetheless, unlike our proposed method, its search strategy for Pareto optimal solutions is entirely random and does not ensure adequate coverage of the Pareto front. Moreover, DGEMO requires a specialized data structure that must be designed separately for each specific number of objectives, thereby limiting its scalability beyond problems with three objectives. Recently, the HVKG [12] acquisition function, which extends the SO Knowledge Gradient acquisition function, has been proposed to tackle multi-fidelity and decoupled MOO problems. Although our proposed method uses the Hypervolume Improvement acquisition function, it is compatible with alternative MO acquisition functions, rendering it applicable to new developments. In the experiment section, we compare against JES - a state-of-the-art IT-based method, as well as qEHVI and DGEMO, as they also leverage the HV-based acquisition functions.

Evolutionary Algorithms (EA) are also capable of tackling MOO problems. Examples of multi-objective Evolutionary Algorithms include MOEA/D [57], SMS-EMOA [6], and NSGA-II [16, 26]. Generally, EA-based algorithms are less sample-efficient than BO-based methods, making them unsuitable for settings with a limited evaluation budget. We compare against NSGA-II, the most widely used EA baseline in MOBO works.

## 4 Proposed Method

In this section, we present our proposed algorithm MOBO-OSD. See Fig. 1 for a summary of the proposed MOBO-OSD method. The core idea of MOBO-OSD is to find the Pareto optimal solutions by estimating the intersection points between the boundary of the objective space and the vectors orthogonal to the convex hull of individual objective minima (CHIM) [11]. To achieve this in the MOBO setting with limited evaluation budget, we first approximate the CHIM using a bounded hyperplane (the *approximated CHIM*), and then construct a set of well-distributed orthogonal search directions (OSDs) w.r.t. this approximated CHIM (Sec. 4.1). Then we propose the *MOBO-OSD subproblem* for each OSD (Sec. 4.2), followed by the incorporation of the Pareto Front Estimation technique (Sec. 4.3). Finally, we propose the batch selection process (Sec. 4.4). For simplicity, we assume that all the objective values are non-negative, which can be achieved by offsetting the current worst value found so far in each objective.

## 4.1 The MOBO-OSD Components

**Approximated CHIM.** As it is not possible to obtain the individual minima of the objectives in MOBO settings with a limited evaluation budget, we propose to approximate the CHIM as a convex hull of $M$ *boundary points*, which are the extreme points computed from the observed dataset. Denote the ideal point $\mathbf{y}^{\text{ideal}} = (\min_{\mathbf{x} \in \mathcal{D}} f_1(\mathbf{x}), \dots, \min_{\mathbf{x} \in \mathcal{D}} f_M(\mathbf{x}))$ and the nadir point $\mathbf{y}^{\text{nadir}} = (\max_{\mathbf{x} \in \mathcal{D}} f_1(\mathbf{x}), \dots, \max_{\mathbf{x} \in \mathcal{D}} f_M(\mathbf{x}))$ as $M$-dimensional points whose components are the best and the worst values observed so far for each objective, respectively. Then, the $m$-th boundary point $\mathbf{p}_m$ is computed by replacing the $m$-th component of the ideal point with its corresponding nadir value, leaving the rest unchanged, i.e., $\mathbf{p}_m$ satisfies $[\mathbf{p}_m]_j = [\mathbf{y}^{\text{nadir}}]_j$ if $j = m$ and $[\mathbf{y}^{\text{ideal}}]_j$ otherwise, for $j = 1, \dots, M$. This approximated CHIM serves as a replacement for the true CHIM, which is not available as individual

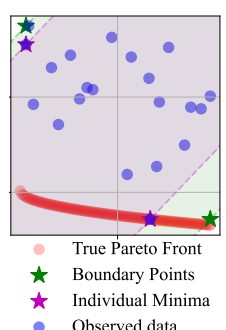

Legend:
- True Pareto Front
- Boundary Points
- Individual Minima
- Observed data

minima cannot be found efficiently in MOBO settings. Compared to a simpler alternative, which approximates the CHIM via individual minima found so far in the observed dataset, our approach avoids prematurely shrinking the search region for the Pareto optimal solutions. In the alternative method, if a good Pareto optimal solution is found early, the individual minima found so far in the observed dataset immediately shrink the search region, overlooking other potential unexplored regions of the objective space to find more Pareto optimal solutions. In contrast, the proposed approximated CHIM maintains a broader search region, allowing our proposed method MOBO-OSD to explore more promising regions. See the inset for an example where our proposed approximated CHIM provides a larger search region (green area) than that of the alternative method (purple area).

**Orthogonal Search Directions.** Having obtained the approximated CHIM, the Orthogonal Search Direction (OSD) is defined as a one-dimensional line in the objective space that follows the unit normal vector $\mathbf{n}$ of the approximated CHIM and passes through a point on the approximated CHIM. Denote $\boldsymbol{\beta}$ as an $M$-dimensional convex combination vector $\boldsymbol{\beta} \in \mathbb{R}^M, \sum_{m=1}^{M} \beta_m = 1$ and $\beta_m > 0$, which defines a point on the approximated CHIM $\mathcal{U}(\boldsymbol{\beta})$. Formally, $\mathcal{U}(\boldsymbol{\beta}) = \{\mathbf{P}\boldsymbol{\beta} = \sum_{m=1}^{M} \beta_m \mathbf{p}_m \mid \boldsymbol{\beta} \in \mathbb{R}^M, \sum_{m=1}^{M} \beta_m = 1, \beta_m > 0\}$ such that $\mathbf{P} = [\mathbf{p}_1, \dots, \mathbf{p}_M]$ is the matrix whose columns are the $M$ boundary points $\mathbf{p}_m$. We then denote the OSD as a line $\mathcal{L}(\mathcal{U}(\boldsymbol{\beta}), \mathbf{n})$ that goes through the point $\mathcal{U}(\boldsymbol{\beta})$ in the direction of the normal vector $\mathbf{n}$. In order to generate a well-distributed set of OSDs, it is essential to construct a well-distributed set of $\mathcal{U}(\boldsymbol{\beta})$ points. This is equivalent to generating a well-distributed set of $\{\boldsymbol{\beta}\}$ over an $M$-dimensional unit simplex, as $\mathcal{U}(\boldsymbol{\beta})$ is, by definition, the linear transformation of $\boldsymbol{\beta}$ from a unit simplex to the approximated CHIM. To achieve this, we propose to employ the Riesz s-Energy method [8] to arrange the set of $\boldsymbol{\beta}$ in a well-distributed fashion. Following the physics principle that the minimum potential energy state of a set of points corresponds to a diverse distribution of those points, the Riesz s-Energy aims to minimize the sum of a potential energy function defined over $\{\boldsymbol{\beta}\}$, which effectively generates a well-distributed set of $\boldsymbol{\beta}$. To sum up, in practice, we generate $n_\beta$ well-distributed OSDs by first producing $n_\beta$ well-distributed convex combination weight vectors $\{\boldsymbol{\beta}_i\}$, then linearly mapping them to the corresponding points $\mathcal{U}(\boldsymbol{\beta}_i)$ on the approximated CHIM, and finally defining the set of OSDs $\{\mathcal{L}(\mathcal{U}(\boldsymbol{\beta}_i), \mathbf{n})\}$, where $i = 1, \dots, n_\beta$. See Fig. 1b for an illustration of the OSDs. More details on the OSDs are in Appendix A.3.

## 4.2 The MOBO-OSD Subproblem

We now present the MOBO-OSD subproblem, which is a constrained optimization problem designed to obtain the Pareto optimal solutions by finding the intersection between an OSD and the boundary of the objective space. Given a convex combination weight vector $\boldsymbol{\beta}$, a point on the approximated CHIM $\mathcal{U}(\boldsymbol{\beta})$, and an OSD $\mathcal{L} = \mathcal{L}(\mathcal{U}(\boldsymbol{\beta}), \mathbf{n})$, the intersection point $\mathbf{L}^*$ between the line $\mathcal{L}$ and the boundary of the objective space can be parameterized as $\mathbf{L}^* = \mathcal{U}(\boldsymbol{\beta}) + \lambda^* \mathbf{n}$, where the scalar parameter $\lambda^*$ satisfies $(\mathbf{x}^*, \lambda^*) \in \max_{\lambda \in \mathbb{R}, \mathbf{x} \in \mathcal{X}} \lambda$ subject to $\mathcal{U}(\boldsymbol{\beta}) + \lambda \mathbf{n} = \mathbf{f}(\mathbf{x})$. This is the ideal optimization subproblem used in the NBI method. More details on the NBI technique and its subproblem can be found in Appendix A.1. In MOBO settings, however, it is not feasible to solve this ideal constrained problem due to the high evaluation cost of the constraints. Therefore, we propose the MOBO-OSD subproblem, which aims to approximate $\lambda^*$

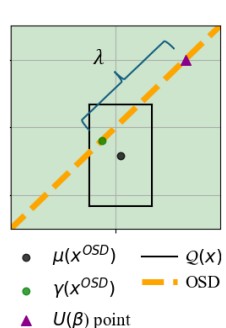

Legend:
- $\mu(x^{OSD})$    — $\mathcal{Q}(x)$
- $\gamma(x^{OSD})$    ⁃⁃ OSD
- $\mathcal{U}(\beta)$ point

and $\mathbf{x}^*$. We modify the ideal problem as follows. First, we replace the objective function values $\mathbf{f}(\mathbf{x})$ with the estimated function values, the posterior mean vector $\boldsymbol{\mu}(\mathbf{x})$. Second, we relax the equality constraint that forces $\boldsymbol{\mu}(\mathbf{x})$ to be exactly on the OSD, $\mathcal{U}(\boldsymbol{\beta}) + \lambda\mathbf{n} = \boldsymbol{\mu}(\mathbf{x})$, and only require $\boldsymbol{\mu}(\mathbf{x})$ to be within a certain distance from the OSD. This can be achieved by constraining the projection of $\boldsymbol{\mu}(\mathbf{x})$ within its confidence bounds, i.e., $\gamma(\mathbf{x}; \boldsymbol{\beta}, \mathbf{n}) \in \mathcal{Q}(\mathbf{x})$ where $\gamma(\mathbf{x}; \boldsymbol{\beta}, \mathbf{n})$ is the projection function of $\boldsymbol{\mu}(\mathbf{x})$ onto the OSD $\mathcal{L}$, and $\mathcal{Q}(\mathbf{x}) = \{\mathbf{y} \mid \boldsymbol{\mu}(\mathbf{x}) - \delta\boldsymbol{\sigma}(\mathbf{x}) \leq \mathbf{y} \leq \boldsymbol{\mu}(\mathbf{x}) + \delta\boldsymbol{\sigma}(\mathbf{x})\}$ represents a hyper-rectangular confidence region defined by the posterior standard deviation vector $\boldsymbol{\sigma}$ and a scaling factor $\delta$ [61]. Finally, we maximize the scalar $\lambda$, which represents the scalar parameter corresponding to the projection point on the OSD $\mathcal{L}$, i.e., $\gamma(\mathbf{x}; \boldsymbol{\beta}, \mathbf{n}) = \mathcal{U}(\boldsymbol{\beta}) + \lambda\mathbf{n}$. See the inset for an illustration of the MOBO-OSD subproblem. Note that $\lambda$ is not necessarily the distance $\|\gamma(\mathbf{x}) - \mathcal{U}(\boldsymbol{\beta})\|_2$, as $\lambda$ can be negative in the case of a concave Pareto front, e.g., the DTLZ2 benchmark function [17]. Ultimately, the MOBO-OSD subproblem is defined as the following constrained optimization problem,

$$(\mathbf{x}^{\mathrm{OSD}}, \lambda^{\mathrm{OSD}}) \in \max_{\mathbf{x} \in \mathcal{X}} \lambda \quad \text{subject to} \quad \begin{cases} \mathbf{g}_1(\mathbf{x}) = \gamma(\mathbf{x}; \boldsymbol{\beta}, \mathbf{n}) - \boldsymbol{\mu}(\mathbf{x}) + \delta\boldsymbol{\sigma}(\mathbf{x}) \geq \mathbf{0} \\ \mathbf{g}_2(\mathbf{x}) = \boldsymbol{\mu}(\mathbf{x}) + \delta\boldsymbol{\sigma}(\mathbf{x}) - \gamma(\mathbf{x}; \boldsymbol{\beta}, \mathbf{n}) \geq \mathbf{0} \end{cases}. \quad (2)$$

A detailed formula for $\lambda$ can be derived from the definition of the projection operator, i.e., $\gamma(\mathbf{x}; \boldsymbol{\beta}, \mathbf{n}) = \mathrm{Proj}_{\mathcal{L}}(\boldsymbol{\mu}(\mathbf{x})) = \mathrm{Proj}_{\mathbf{n}}(\boldsymbol{\mu}(\mathbf{x}) - \mathcal{U}(\boldsymbol{\beta})) = \mathcal{U}(\boldsymbol{\beta}) + \frac{(\boldsymbol{\mu}(\mathbf{x}) - \mathcal{U}(\boldsymbol{\beta})) \cdot \mathbf{n}}{\mathbf{n} \cdot \mathbf{n}} \mathbf{n}$. Therefore, $\lambda$ can be expressed as $\lambda(\mathbf{x}; \boldsymbol{\beta}, \mathbf{n}) = (\boldsymbol{\mu}(\mathbf{x}) - \mathcal{U}(\boldsymbol{\beta})) \cdot \mathbf{n}$, as $\|\mathbf{n}\| = 1$. Solving Eq. (2) results in $(\mathbf{x}^{\mathrm{OSD}}, \lambda^{\mathrm{OSD}})$, which approximates the ideal $(\mathbf{x}^*, \lambda^*)$ as Eq. (2) ensures that the obtained solution has competitive estimated function values while remaining close to the current OSD $\mathcal{L}$. The point $\boldsymbol{\mu}(\mathbf{x}^{\mathrm{OSD}})$ serves as an approximation of the intersection point $\mathbf{L}^*$. Given a set of $n_\beta$ well-distributed OSDs as defined in Sec. 4.1, solving a MOBO-OSD subproblem for each OSD $\mathcal{L}$ generates a well-distributed set of solutions $\{\mathbf{x}^{\mathrm{OSD}}\}$. See Fig. 1c for an illustration.

The detailed optimization routine for the proposed MOBO-OSD subproblem in Eq. (2) proceeds as follows. We solve Eq. (2) using a gradient-based off-the-shelf optimizer, e.g., SLSQP [35]. To compute the gradient vector $\mathcal{J}_\lambda(\mathbf{x})$ and Jacobian matrix $\mathcal{J}_{[\mathbf{g}_1, \mathbf{g}_2]}(\mathbf{x})$, we require the Jacobian matrix of the posterior mean and standard deviation vectors of the GPs, which can be computed either in analytic form under common kernels [33, 38] or by automatic differentiation via computational graphs [42]. Moreover, since the problem is highly non-convex, we solve Eq. (2) via multiple starting points $\{\mathbf{x}_i^{(0)}\}_{i=0}^{n_s}$ and select the most promising solution among the resulting solutions. Specifically, for each starting point $\mathbf{x}_i^{(0)}$, we obtain a solution $\mathbf{x}_i^{\mathrm{OSD}}(\boldsymbol{\beta}) = \mathbf{x}_i^{\mathrm{OSD}}$ that exhibits a different trade-off between maximizing $\lambda(\mathbf{x})$ and minimizing the distance from $\mu(\mathbf{x})$ to $\mathcal{L}$, i.e., $l(\mathbf{x}) = \|\mu(\mathbf{x}) - \gamma(\mathbf{x})\|_2$. Since it is not feasible to find the promising solution $\mathbf{x}_i^{\mathrm{OSD}}$ that maximizes $\lambda(\mathbf{x}_i^{\mathrm{OSD}})$ while enforcing $l(\mathbf{x}_i^{\mathrm{OSD}}) = 0$, we opt to select the most potential solution by formulating a bi-objective selection step to identify the best trade-off among the $n_s$ candidates $\{\mathbf{x}_i^{\mathrm{OSD}}\}|_{i=1}^{n_s}$, characterized by $\lambda_i = \lambda(\mathbf{x}_i^{\mathrm{OSD}})$ and $l_i = l(\mathbf{x}_i^{\mathrm{OSD}})$. We use the hypervolume indicator for this selection, computing the hypervolume contribution for each pair $[\lambda_i, l_i]$ as the reduction in hypervolume when that pair is removed from the solution set $\mathbf{S} = \{[\lambda_i, l_i]\}_{i=1}^{n_s}$. Formally, $\mathrm{HVC}([\lambda_i, l_i]) = \mathrm{HV}(\mathbf{S}, \mathbf{r}_s) - \mathrm{HV}(\mathbf{S} \setminus [\lambda_i, l_i], \mathbf{r}_s)$, where $\mathbf{r}_s = \mathbf{s}^{\mathrm{nadir}} + 0.1 \cdot (\mathbf{s}^{\mathrm{nadir}} - \mathbf{s}^{\mathrm{ideal}})$ is the reference point computed from the nadir point $\mathbf{s}^{\mathrm{nadir}}$ and the ideal point $\mathbf{s}^{\mathrm{ideal}}$ of the set $\mathbf{S}$. The final solution for Eq. (2) is chosen as $\mathbf{x}_{i^*}^{\mathrm{OSD}}(\boldsymbol{\beta}) = \mathbf{x}_{i^*}^{\mathrm{OSD}}$, where $i^* = \arg\max_{i=1,\dots,n_s} \mathrm{HVC}([\lambda_i, l_i])$. This procedure ensures the selected solution achieves a balance between a high $\lambda$ value and a close proximity to the line $\mathcal{L}$. Since each MOBO-OSD subproblem in Eq. (2) is solved independently, all subproblems (for different $\boldsymbol{\beta}$) can be processed in parallel to improve computational efficiency.

### 4.3 Pareto Front Estimation

For each MOBO-OSD subproblem (corresponding to a point on the approximated CHIM $\mathcal{U}(\boldsymbol{\beta})$), we obtain a solution $\mathbf{x}^{\mathrm{OSD}}(\boldsymbol{\beta})$ (in Eq. (2)). Instead of discretizing the approximated CHIM into a large number of points $\mathcal{U}(\boldsymbol{\beta})$, we propose to leverage a Pareto Front Estimation (PFE) technique to explore the local region around the current solution $\mathbf{x}^{\mathrm{OSD}}(\boldsymbol{\beta})$ and generate additional data points that are expected to be Pareto optimal. One successful approach is the First Order Approximation technique [46, 33], which computes a local exploration space $\mathcal{T}$ around a current Pareto optimal solution. Specifically, we consider the First Order Approximation problem, in which the surrogate model is used to estimate the exploration space $\mathcal{T}$ around the solution $\mathbf{x}^{\mathrm{OSD}}(\boldsymbol{\beta})$. See Appendix A.2 for an overview of the First Order Approximation method. In particular, solving the First Order Approximation problem results in a set of directions $\mathbf{v}$, which define the space $\mathcal{T}$. Then we estimate

**Algorithm 1** The MOBO-OSD Algorithm
_________________________________________________________________________________
 1: **Input:** Objective function $\mathbf{f}(.)$, evaluation budget $T$, batch size $b$, number of weight vectors $n_\beta$
 2: **Output:** The Pareto set $\mathcal{P}_s$
 3: Initialize data points and append to the observed dataset $\mathcal{D}$
 4: **while** $t \leq T$ **do**
 5:     Compute approximated CHIM and define $n_\beta$ OSDs                               ▷ Sec. 4.1
 6:     Train GPs for each objective function $f_m$
 7:     **for each** point $\mathcal{U}(\boldsymbol{\beta})$ on the approximated CHIM **do**
 8:         Optimize the MOBO-OSD subproblem to generate a candidate $\mathbf{x}^{\text{OSD}}(\boldsymbol{\beta})$     ▷ Eq. (2)
 9:         Estimate the Pareto front around $\mathbf{x}^{\text{OSD}}(\boldsymbol{\beta})$ to explore more candidates $\mathbf{x}^{\text{PFE}}(\boldsymbol{\beta})$ ▷ Eq. (3)
10:         Append $\mathbf{X}_c \leftarrow \mathbf{X}_c \cup \mathbf{x}^{\text{PFE}}(\boldsymbol{\beta})$
11:     **end for**
12:     Select a batch of $b$ solutions from $\mathbf{X}_c$ and evaluate them; Increase $t \leftarrow t + b$       ▷ Eq. (4)
13: **end while**
14: Return $\mathcal{P}_s$ from dataset $\mathcal{D}$
_________________________________________________________________________________

the Pareto Front by randomly sampling $n_e$ data points in $\mathcal{T}$ such that,

$$\mathbf{X}^{\text{PFE}}(\boldsymbol{\beta}) = \{\mathbf{x}_i^{\text{PFE}}(\boldsymbol{\beta})\} = \{\mathbf{x}^{\text{OSD}}(\boldsymbol{\beta}) + \mathbf{u}_i\mathbf{v}\} \text{ for } i = 1, \dots, n_e, \tag{3}$$

where $\mathbf{u}_i$ is the $i$-th random perturbation term to shift the solution $\mathbf{x}^{\text{OSD}}$ along the directions $\mathbf{v}$. This PFE step generates additional candidate points around the current expected Pareto optimal candidates $\mathbf{x}^{\text{OSD}}$ from the previous step. This also helps to avoid solving an excessive number of MOBO-OSD subproblems that are close to one another. See the ablation study in Sec. 5.2 for details.

This step is repeated for all MOBO-OSD subproblems, i.e., for each point on the approximated CHIM $\mathcal{U}(\boldsymbol{\beta})$, resulting in a set of candidates $\mathbf{X}_c = \bigcup_{i=1}^{n_\beta} \mathbf{X}^{\text{PFE}}(\boldsymbol{\beta}_i)$, where $n_\beta$ denotes the number of weight vectors $\boldsymbol{\beta}$ on the approximated CHIM. In general, $|\mathbf{X}_c| = n_\beta \times n_e$.

### 4.4 Batch Selection Strategy

Having generated $n_\beta \times n_e$ candidate solutions $\mathbf{X}_c$, we use the Hypervolume Improvement (HVI) of the posterior mean as the acquisition function to determine the most promising solutions to be evaluated - solutions that are expected to maximize the hypervolume contribution [9, 33]. Given the current approximate Pareto front $\mathcal{P}_f$ computed from the observed dataset $\mathcal{D}$ and a reference point $\mathbf{r}$, the HVI acquisition function is defined as $\alpha_{\text{HVI}}(\mathbf{x}; \mathcal{P}_f, \mathbf{r}) = \text{HV}(\boldsymbol{\mu}(\mathbf{x}) \cup \mathcal{P}_f, \mathbf{r}) - \text{HV}(\mathcal{P}_f, \mathbf{r})$, where $\boldsymbol{\mu}(.)$ is the posterior mean function. For batch selection, the goal is to pick a batch of $b$ data points $\mathbf{X}_b$ that achieve high HVI values while maintaining diversity among the chosen solutions. To address the problem, first, we apply the Kriging Believer (KB) method [22] to improve the posterior mean estimation given unobserved data points during batch selection. In particular, after selecting a data point $\mathbf{x}_i$ for $i \in [1, \dots, b]$, we re-train the GP models on the aggregated dataset $\mathcal{D} \cup \{\mathbf{x}_i, \boldsymbol{\mu}(\mathbf{x}_i)\}$. Secondly, we boost the diversity by choosing candidate points coming from different exploration spaces $\mathcal{T}$. This is because the approximated candidate points from a single exploration space $\mathcal{T}$ tend to be close to one another [46], therefore exhibiting similar contributions to the hypervolume. Denote $\psi(\boldsymbol{\beta}_i, \mathbf{X})$ as a function that counts the number of candidates $\mathbf{x} \in \mathbf{X}$ originating from the exploration space $\mathcal{T}$ around $\mathbf{x}^{\text{OSD}}(\boldsymbol{\beta}_i)$. Hence, the batch selection mechanism is formulated as follows,

$$\mathbf{X}_b = \underset{\mathbf{X}_b \in \mathbf{X}_c}{\arg\max} \, \alpha_{\text{HVI}}(\mathbf{x}; \mathcal{P}_f, \mathbf{r}) \quad \text{such that} \quad \max_{i=1,\dots,n_\beta} \psi(\boldsymbol{\beta}_i, \mathbf{X}_b) - \min_{i=1,\dots,n_\beta} \psi(\boldsymbol{\beta}_i, \mathbf{X}_b) \leq 1. \tag{4}$$

The constraints in Eq. (4) are designed to ensure that the number of data points selected from different exploration spaces differs by at most one. In practice, we solve Eq. (4) sequentially to select $b$ data points $\mathbf{X}_b$ from the candidate set $\mathbf{X}_c$. In each iteration, after re-training the GP models (as required by KB) and selecting the best candidate point based on $\alpha_{\text{HVI}}$, we remove from $\mathbf{X}_c$ all other candidates originating from the same exploration space, then repeat the process to select the next point in the batch. If the candidate set $\mathbf{X}_c$ becomes empty, all previously removed but unselected candidate points are reintroduced, and the process is repeated.

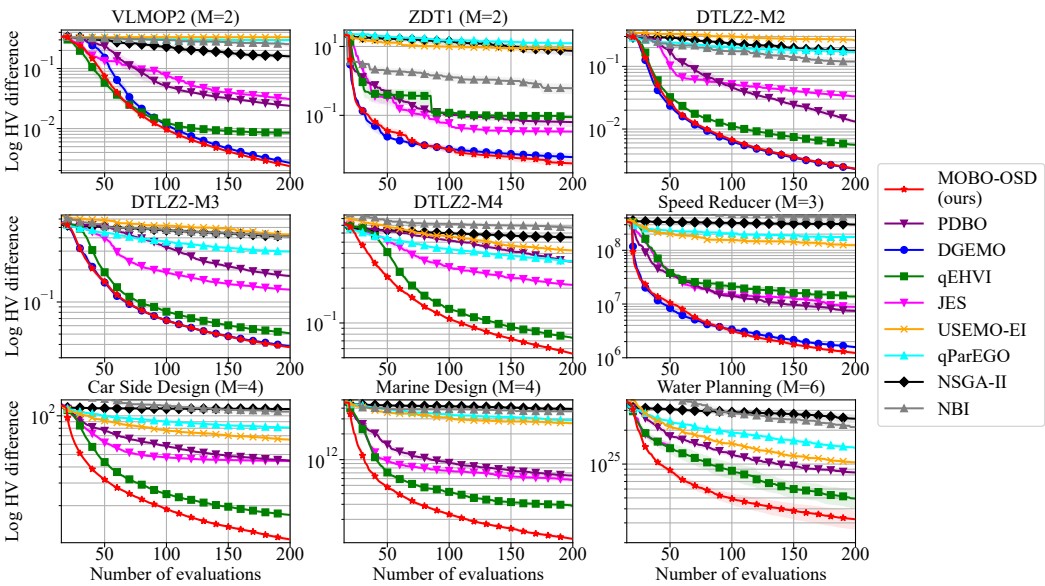

Figure 2: Comparison of MOBO-OSD against the SOTA baselines on 5 synthetic and 4 real-world benchmark problems in sequential setting (batch size 1). Note that DGEMO does support problems with $M > 3$ objectives. Overall, MOBO-OSD outperforms the baselines.

## 4.5 Overall Algorithm

The overall MOBO-OSD algorithm is described in Alg. 1. MOBO-OSD operates in an iterative fashion. In each iteration, based on the observed dataset $\mathcal{D}$, we compute $M$ boundary points $\mathbf{P}$, the approximated CHIM $\mathcal{U}$, and the $n_\beta$ OSDs (line 5). Subsequently, for each point $\mathcal{U}(\boldsymbol{\beta})$ on the approximated CHIM, we optimize the MOBO-OSD subproblem to generate a Pareto optimal candidate $\mathbf{x}^{\text{OSD}}$, and then locally explore the space around $\mathbf{x}^{\text{OSD}}$ to generate additional Pareto optimal candidates $\mathbf{x}^{\text{PFE}}$ (lines 8 - 9). We then use the aggregated set of candidates $\mathbf{X}_c$ from all $n_\beta$ OSDs to select a batch of $b$ solutions using the $\alpha_{\text{HVI}}$ acquisition function and the exploration space constraint (line 12). The process is repeated until the evaluation budget is exhausted, and the final Pareto front $\mathcal{P}_f$ and Pareto set $\mathcal{P}_s$ are computed from the observed dataset $\mathcal{D}$.

## 5 Experiments

Now we empirically evaluate our proposed method, MOBO-OSD, against the state-of-the-art methods on an extensive set of synthetic and real-world benchmark problems.

**Experimental Settings and Baselines.** We evaluate the proposed algorithm **MOBO-OSD** against a comprehensive set of baselines: **qParEGO** [31], **USeMO** [5], **DGEMO** [33], **PDBO** [1], **JES** [52], **qEHVI** [13], **NSGA-II** [16] and **NBI** [11]. For USeMO, we select the EI acquisition function as it has good performance and is commonly used in other works. For qParEGO, we use the batch implementation developed by Daulton et al. [13]. For DGEMO, we note that the authors' implementation only supports problems with $M = 2$ and $M = 3$ objectives. For JES, due to its prohibitive computational cost (Appendix A.11), we only compare under the sequential optimization setting (batch size 1). For NBI, to the best of our knowledge, no open-source implementation of the method currently exists; therefore, we re-implemented it following the procedure described in [11]. Detailed implementations of MOBO-OSD and the baselines can be found in Appendix A.7.

For the comparison metrics, we compute the logarithmic hypervolume difference between the hypervolume of the best accumulated observed Pareto front and the maximum hypervolume. The hypervolume is calculated using the reference points $\mathbf{r}$ specified in Appendix A.6. We report the mean and the standard error across 10 independent runs.

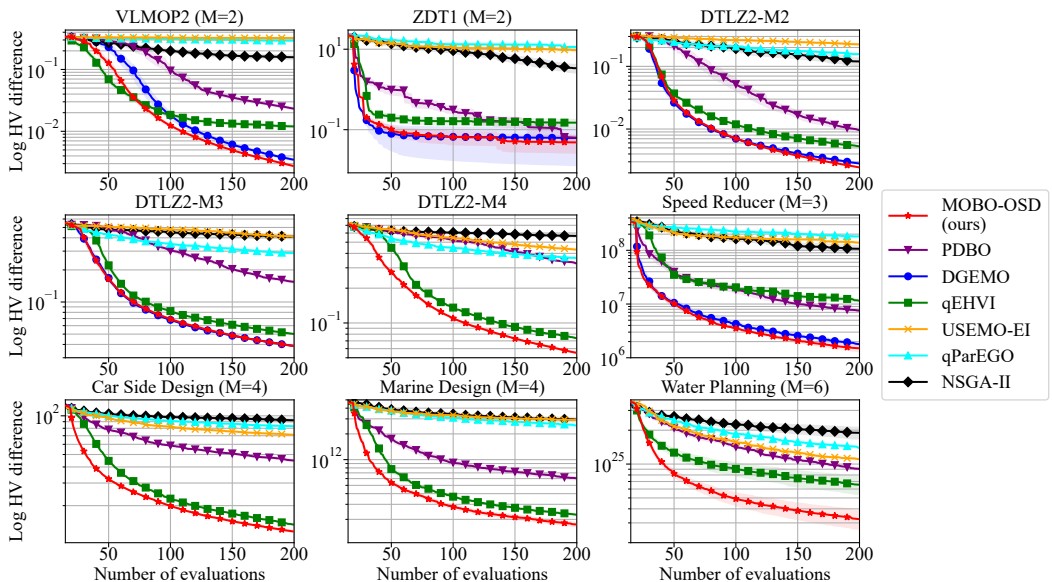

Figure 3: Comparison of MOBO-OSD against the SOTA baselines on 5 synthetic and 4 real-world benchmark problems in batch setting (batch size 4). Overall, MOBO-OSD outperform the baselines.

**Benchmark Problems.** We conduct experiments on five synthetic and four real-world multi-objective benchmark problems. The number of objectives ranges from two to six, which is common in the MOBO literature. For synthetic benchmark problems, we use *DTLZ2* with different objective settings $M \in \{2, 3, 4\}$ [17], *ZDT1* [15], and *VLMOP2* [53]. For real-world benchmark problems, we use various problems from the RE problem suite [50] including *Speed Reducer*, *Car Side Design*, *Marine Design*, and *Water Planning*. The dimensionality and number of objective settings for each function are given in Table 2. These problems are widely used in the MOBO literature [5, 9, 13, 12, 1, 33]. Details of the benchmark problems can be found in Appendix A.6.

## 5.1 Comparison with Baselines

**Sequential Optimization.** Fig. 2 shows the performance of all baselines on all nine benchmark problems in the sequential setting (batch size 1). Across all benchmark problems, both synthetic and real-world, MOBO-OSD consistently outperforms other state-of-the-art methods. qEHVI shows competitive performance in most cases, yet eventually finds suboptimal solutions. DGEMO also shows strong performance, but is outperformed by MOBO-OSD on VLMOP2 and Speed Reducer. Moreover, it is limited to problems with at most three objectives $M \leq 3$. NSGA-II often cannot compete with the MOBO algorithms in the case of limited evaluation budget. These results indicate the efficiency of MOBO-OSD in achieving fast convergence towards the Pareto front.

**Batch Optimization.** We conduct experiments in batch settings, with batch size $b = \{4, 8, 10\}$. Fig. 3 shows the performance of all baselines across all nine benchmark problems with batch size 4, whereas additional batch results can be found in Appendix A.8. Similar to the batch size 1 setting, MOBO-OSD consistently outperforms other state-of-the-art methods. qEHVI and DGEMO remain the two strongest baselines after MOBO-OSD, achieving hypervolume results close to those of MOBO-OSD. This result further emphasizes the efficiency of MOBO-OSD, even in the batch setting. Furthermore, we provide a theoretical time complexity analysis for MOBO-OSD, along with a runtime comparison between MOBO-OSD and the baselines in Appendix A.11, demonstrating the scalability of MOBO-OSD to an arbitrary number of objectives in both sequential and batch settings.

## 5.2 Ablation Study

In this section, we conduct a study on the effect of the $n_\beta$ parameter - the number of points on the approximate CHIM $\mathcal{U}(\boldsymbol{\beta})$ - on the performance of MOBO-OSD. A larger $n_\beta$ corresponds to a denser set of points on the approximated CHIM. We evaluate MOBO-OSD using varying

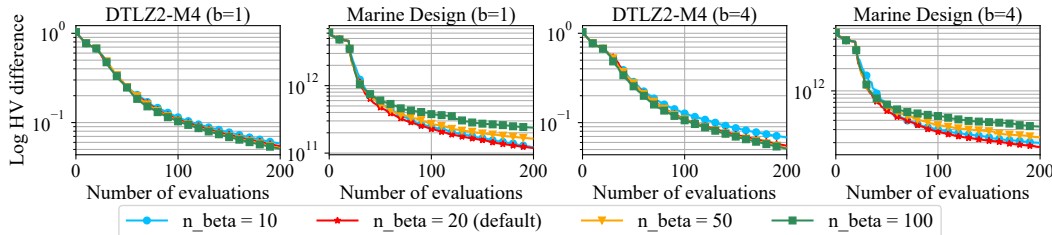

Figure 4: Ablation study on the effect of parameter $n_\beta$. Full results can be found in the Appendix A.9. Overall, MOBO-OSD is robust against $n_\beta$.

values of $n_\beta \in \{10, 50, 100\}$ on all nine benchmark problems and compare to the default setting ($n_\beta = 20$). Four representative results are shown in Fig. 4, while the remaining results can be found in Appendix A.9. Fig. 4 shows that MOBO-OSD's overall performance remains stable across different values of $n_\beta$, indicating the robustness of the proposed algorithm with respect to this parameter.

We hypothesize that this robustness can be attributed to the Pareto Front Estimation (PFE) component (Sec. 4.3), which facilitates the generation of additional Pareto optimal candidates. Without the component, MOBO-OSD would require more $\mathcal{U}(\boldsymbol{\beta})$ points, thus solving a larger number of MOBO-OSD subproblems to achieve a comparable set of Pareto optimal candidates. To validate this hypothesis, we conduct a study by removing the PFE component and run the resulting MOBO-OSD variant with varying values of $n_\beta \in \{20, 100, 200, 500\}$. As shown in Table 1, increasing $n_\beta$ leads to performance improvements. This finding suggests that although PFE enhances the efficiency of MOBO-OSD, it is not an essential component, as similar performance can be attained by increasing the density of the $\mathcal{U}(\boldsymbol{\beta})$ set.

Table 1: Ablation results (in HV) on the effect of $n_\beta$ on MOBO-OSD without the PFE component.

| MOBO-OSD method | DTLZ2-M2 | VLMOP2 | Car Side Design |
|---|---|---|---|
| W/o PFE ($n_\beta = 20$) | $0.4041 \pm 0.0004$ | $0.2713 \pm 0.0020$ | $145.1195 \pm 0.3340$ |
| W/o PFE ($n_\beta = 100$) | $0.4118 \pm 0.0001$ | $0.2978 \pm 0.0011$ | $154.3249 \pm 0.2662$ |
| W/o PFE ($n_\beta = 200$) | $0.4142 \pm 0.0001$ | $0.3076 \pm 0.0006$ | $157.2311 \pm 0.2095$ |
| W/o PFE ($n_\beta = 500$) | $0.4164 \pm 0.0001$ | $0.3159 \pm 0.0004$ | $160.2797 \pm 0.2118$ |
| **Default (with PFE)** | $0.4217 \pm 0.0000$ | $0.3383 \pm 0.0000$ | $177.4782 \pm 0.2310$ |

## 6   Conclusions

In this paper, we address the multi-objective Bayesian optimization problem for expensive black-box, vector-valued objective functions. We propose MOBO-OSD, a novel algorithm that aims to generate a well-distributed set of solutions via multiple subproblems defined along orthogonal search directions. To further enrich the diversity of the solutions, we perform local exploration around current Pareto optimal candidates, generating additional Pareto optimal candidate solutions. These candidates are scored using the Hypervolume Improvement acquisition function, while batch selection is guided by the Kriging Believer strategy and the exploration space information. Our experimental results show that MOBO-OSD outperforms state-of-the-art methods across various synthetic and real-world benchmark problems with varying number of objectives, in both sequential and batch settings.

**Limitations.**   One limitation of our approach is that it focuses on noiseless observations, a common assumption in various existing works. Future work could address this limitation by employing acquisition functions that can handle noise and by improving the MOBO-OSD subproblem - for example, by integrating the uncertainty of previous observations when defining the projection.

## Acknowledgments

The first author (L.N.) would like to thank the School of Computing Technologies, RMIT University, Australia and the Google Cloud Research Credits Program for providing computing resources for this project. Additionally, this project was undertaken with the assistance of computing resources from RMIT Advanced Cloud Ecosystem (RACE).

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

# A Technical Appendices and Supplementary Material

## A.1 Normal Boundary Intersection

In this section, we provide more details on the Normal Boundary Intersection (NBI) [11] method, which is a general MOO technique that aims to generate a uniformly distributed set of Pareto optimal solutions. The key insight of the NBI method is that, given a MOO problem, the intersection points between the boundary of the objective space and the vectors orthogonal to the convex hull of individual minima (CHIM) of the objectives could be Pareto optimal solutions. Given the MOO problem in Eq. (1), the NBI method first determines the individual optima of each objective $\mathbf{f}_m^* = \mathbf{f}(\mathbf{x}_m^*)$ where $\mathbf{x}_m^* = \arg\min_{\mathbf{x} \in \mathcal{X}} f_m(\mathbf{x})$ for $m = 1, \ldots, M$. Denote a column matrix $\mathbf{F}^* = [\mathbf{f}_1^*, \mathbf{f}_2^*, \ldots, \mathbf{f}_M^*]^\mathsf{T}$, the CHIM is constructed by a set of points $\{\mathcal{U}(\boldsymbol{\beta})\}$ such that $\mathcal{U}(\boldsymbol{\beta}) = \{\mathbf{F}^*\boldsymbol{\beta} = \sum_{i=1}^{M} \beta_i \mathbf{f}_i \mid \boldsymbol{\beta} \in \mathbb{R}^M, \sum_{i=1}^{M} = 1, \beta_i > 0\}$, where $\boldsymbol{\beta} = [\beta_1, \ldots, \beta_M]$ is a convex combination weight vector to construct points on CHIM. For each point $\mathcal{U}(\boldsymbol{\beta})$, NBI defines a constrained optimization problem, referred to as the NBI subproblems, along the direction normal to CHIM, i.e., $(\mathbf{x}^*, \lambda^*) \in \max_{\lambda \in \mathbb{R}, \mathbf{x} \in \mathcal{X}} \lambda$ subject to $\mathcal{U}(\boldsymbol{\beta}) + \lambda \mathbf{n} = \mathbf{f}(\mathbf{x})$, where $\mathbf{n}$ is the normal vector of CHIM. Solving this constrained optimization problem results in a data point $\mathbf{x}^*$ corresponding to the intersection point $\mathbf{f}(\mathbf{x}^*)$ that is expected to be Pareto optimal. As a result, by solving multiple NBI subproblems defined over a set of well-distributed points $\mathcal{U}(\boldsymbol{\beta})$ on the CHIM, the NBI method is expected to result in a set of well-distributed $n_\beta$ Pareto optimal solutions for the MOO problem in Eq. (1). The geometric intuition ensures uniform spacing of solutions and is particularly suitable for problems with complex, non-convex Pareto fronts. However, solving the NBI problem requires evaluating the objective functions $\mathbf{f}(\mathbf{x})$ for the constraint term, which is impractical for expensive objective functions and limited evaluation budget as in MOBO.

In NBI, the set of well-distributed $\boldsymbol{\beta} = [\beta_1, \ldots, \beta_M]$ is constructed using a structured approach as follows. Given an integer number denoting the number of partition $n_p$, possible values for each $\beta_i$ are $\{0, \delta, 2\delta, \ldots, 1\}$ where $\delta = 1/n_p$ is the step-size. Then, we sequentially select each $\beta_i$ such that they satisfy $\sum_{i=1}^{M} \beta_i = 1$. In practice, the possible values for $\beta_j$ corresponding to $\beta_i = m_i \delta$ for $i = 1, \ldots, j-1$ and $j = 2, \ldots, M-1$ are $\left\{0, \delta, 2\delta, \ldots, (p - \sum_{i=1}^{j-1} m_i)\delta\right\}$. Because of this formulation, the number of convex combination weight vectors $\boldsymbol{\beta}$ (and the number of points on CHIM $\mathcal{U}(\boldsymbol{\beta})$) is determined via the binominal coefficients as $n_\beta = \binom{M+p-1}{p}$. As a result, this structured formulation cannot generate *any arbitrary number of* $n_\beta$, as it ultimately depends on the number of partitions $p$ instead. Our proposed MOBO-OSD algorithm leverages a different point generation method (Sec 4.1) that can handle any arbitrary number of $n_\beta$, while maintaining the desired well-distributed property.

## A.2 Pareto Front Estimation via First Order Approximation

This section summarizes the First Order Approximation technique [46] to estimate the Pareto front of a MO problem. In MOO, discovering the entire Pareto front $\mathcal{P}_f$ can be challenging, however, once we find a Pareto optimal solution $\mathbf{x}^* \in \mathcal{P}_f$, it is easier to find nearby Pareto optimal solutions by locally exploring around $\mathbf{x}^*$ [46, 33]. Pareto Front First Order Approximation technique [46] leverages this fact to approximate the Pareto front from a previously discovered Pareto optimal solution. Given the MOO problem in Eq. (1) and an observed Pareto optimal input $\mathbf{x}^*$ found so far, the key idea is to construct a local exploration space $\mathcal{T} \subset \mathcal{X}$ around $\mathbf{x}^*$. In particular, $\mathcal{T}$ is a linear combination of a set of exploration vectors $\mathbf{v}$ which are obtained by solving the following equation:

$$\begin{cases} \mathbf{H}\mathbf{v} \in \mathrm{Im}\left(\mathcal{J}_F(\mathbf{x}^*)\right) \; \oplus \; \mathrm{Im}\left(\mathcal{J}_G(\mathbf{x}^*)\right), \\ \mathcal{J}_G^\mathsf{T}(\mathbf{x}^*)\mathbf{v} = 0, \end{cases} \tag{5}$$

where $\mathbf{H} = \sum_{i=1}^{M} \alpha_i \mathcal{H}_{f_i}(\mathbf{x}^*) + \sum_{k=1}^{K} \beta_k \mathcal{H}_{g_k}(\mathbf{x}^*)$ corresponds to the derivatives of the stationarity KTT condition; $\mathcal{J}_F$ and $\mathcal{J}_G$ are the Jacobian matrix of all objectives $\mathbf{f}$ and $K$ active constraints $\mathbf{g}$, respectively; $\mathcal{H}_u$ is the Hessian matrix of an arbitrary function $u$ and $\boldsymbol{\alpha} = [\alpha_1, \ldots, \alpha_M]$ and $\boldsymbol{\beta} = [\beta_1, \ldots, \beta_K]$ are the Lagrange multipliers (dual variables) associated with the KKT conditions [24] of the MOO problem in Eq. (1) . The first equation in Eq. (5) ensures that moving $\mathbf{x}^*$ along direction $\mathbf{v}$ still preserves the stationarity KKT conditions, ensuring the Pareto optimality of the resulting solutions. Additionally, the second equation ensures the feasibility of active constraints $\mathbf{g}$ that appear when $\mathbf{x}^*$ is very close to a boundary of $\mathcal{X}$. Overall, this technique can help to discover more

solutions nearby the current Pareto optimal solution $\mathbf{f}(\mathbf{x}^*)$. Schulz et al. [46] shows that, generally, $\dim(\mathcal{T}) = \min(M - 1, D)$. Having the local exploration space $\mathcal{T}$ with exploration vectors $\mathbf{v}$, we can generate additional Pareto optimal solutions around the previously discovered Pareto optimal solutions $\mathbf{x}^*$.

### A.3 Details on the Proposed Orthogonal Search Directions

This section presents the details when computing the normal direction of the approximated CHIM (Sec. 4.1). In MOBO-OSD, following Das and Dennis [11], instead of using the exact normal direction $\bar{\mathbf{n}}$ where $\mathbf{P}\bar{\mathbf{n}}^{\mathsf{T}} = \mathbf{0}$, we employ quasi-normal directions for consistent scaling across all objectives, preventing potential ill-conditioning. Specifically, the proposed orthogonal search direction $\hat{\mathbf{n}}$ for MOBO-OSD is defined as $\hat{\mathbf{n}} = -\mathbf{P}e$ where $e$ is a column vector of all ones. The formula computes an equally weighted linear combination of the boundary points $\mathbf{p}_m$, then multiplied by -1 to ensure the normal vector points to the origin. Then we normalize normal vector to unit length $\mathbf{n} = \hat{\mathbf{n}}/\|\hat{\mathbf{n}}\|$. The quasi-normal direction can be interpreted as applying a normalization to remove the difference in scaling among the objectives, while maintaining similar results of subproblems, i.e., the intersection points found.

### A.4 Advantages of OSD Subproblems Compared to Other Scalarization Techniques

In this section, we provide a discussion on the advantages of our propose OSD subproblem formulation compared to the most closely related scalarization technique - linear scalarization (LS) [41]. First, *LS generates search directions at random*, which can be less efficient than the data-driven search directions proposed in our OSD formulation. Second, even when LS weight vectors are well-distributed (e.g., using Riesz s-Energy [8]), *LS is limited to finding solutions on the convex regions of the Pareto front* [58]. In contrast, our proposed OSD formulation is capable of identifying solutions on the Pareto front of arbitrary shape, including both convex and non-convex regions. Note that due to this limitation, LS has been superseded by the most widely used scalarization technique - Tchebychev scalarization (TCH) [41]. TCH can handle non-convex PF and has been widely selected in other works when evaluating scalarization techniques [41, 31, 13, 33]. However, one critical drawback of TCH lies in its non-smooth formulation caused by the maximization operator, which makes TCH suffers from non-differentiability and slow convergence [36]. On the other hand, the MOBO-OSD subproblem formulation is differentiable, either analytically under common kernels or by automatic differentiation via computational graphs, as described in Sec. 4.2. Empirically, our proposed method consistently outperforms baselines employing TCH, such as qParEGO. Figures 2, 3, 5 and 6 show that MOBO-OSD outperforms qParEGO across different benchmark functions with diverse PF characteristics, including convex (ZDT1) and concave (DTLZ2).

### A.5 Comparison To Other Line-based BO Algorithms

As our proposed MOBO-OSD algorithm conducts search for Pareto optimal solution along *one-dimensional guiding lines*, in this section, we discuss the key differences compared to other recent line-based BO algorithms, such as LineBO [30] and BOIDS [40]. The primary distinction lies in the formulation of these one-dimensional lines: both LineBO and BOIDS construct search directions with some degree of *randomness*, whereas MOBO-OSD relies on *deterministic* OSD directions that are orthogonal to the approximated CHIM. Additionally, another key difference is in the search domain: LineBO and BOIDS define one-dimensional lines in the *input space*, while our MOBO-OSD formulates one-dimensional search directions in the *output space*. Furthermore, LineBO and BOIDS are primarily *designed to make high-dimensional problems tractable*, while MOBO-OSD focuses on *generating a well-distributed set of Pareto optimal points* on the Pareto front. Therefore, LineBO and BOIDS are fundamentally different approaches and are not compatible with the MOBO settings. In contrast, MOBO-OSD is specifically designed for MOBO problems, with the core purpose of generating orthogonal search directions relative to the approximated CHIM to ensure a well-distributed coverage of the Pareto front.

### A.6 Details of Benchmark Problems

We present the details of all benchmark problems in Table 2. The reference points for synthetic problems are similar to many MOBO works [13, 14, 1]. To compute the reference point for real-world

problems, we follow a common rule [13, 33, 50] and generate a pool of observed data points to compute the reference point via the pool's nadir and ideal points as $\mathbf{r} = \mathbf{y}^{\text{nadir}} + 0.1 * (\mathbf{y}^{\text{nadir}} - \mathbf{y}^{\text{ideal}})$. All baselines use similar reference points both when running the code implementation (if required) and when computing the hypervolume difference as comparison metric.

Table 2: Details of 5 synthetic (ZDT1, VLMOP2, DTLZ2-M2, DTLZ2-M3, DTLZ2-M4) and 4 real-world (Speed Reducer, Car Side Design, Marine Design, Water Planning) benchmark problems.

| Problem | D | M | Reference Point |
|---|---|---|---|
| DTLZ2-M2 | 5 | 2 | $(1.1, 1.1)$ |
| DTLZ2-M3 | 5 | 3 | $(1.1, 1.1, 1.1)$ |
| DTLZ2-M4 | 5 | 4 | $(1.1, 1.1, 1.1, 1.1)$ |
| ZDT1 | 5 | 2 | $(11.0, 11.0)$ |
| VLMOP2 | 5 | 2 | $(1.0, 1.0)$ |
| Speed Reducer | 7 | 3 | $(6735.9, 1761.17, 402.34)$ |
| Car Side Design | 7 | 4 | $(38.89, 4.44, 12.94, 8.87)$ |
| Marine Design | 6 | 4 | $(-210.44, 18970.82, 24111.07, 11.36)$ |
| Water Planning | 3 | 6 | $(84349, 1461, 3101484, 12442800, 67030, 1.59)$ |

## A.7 Detailed Implementation

We implemented MOBO-OSD and all baselines in Python (version 3.10). The detailed implementation are as follows.

**MOBO-OSD.** For the surrogate model, we implement the GPs via `GPyTorch` [20] and `BoTorch` [3]. We follow [33] and use Matérn 5/2 kernel with the ARD length-scales in the interval $\sqrt{10^{-3}}, \sqrt{10^3}$ and signal variance in the interval $\sqrt{10^{-3}}, \sqrt{10^3}$. The Gaussian likelihood is modeled with standard homoskedastic noise in the interval $[10^{-6}, 10^{-3}]$.

For the number of points on approximated CHIM, we set the default value $n_\beta = 20$ and present an ablation study of other settings in Sec. 5.2. For the scaling of confidence region, we use the common 95% confidence interval, i.e., $\delta = 1.96$ [21]. For the number of starting points when solving MOBO-OSD subproblem, we set $n_s = 4$. Our code implementation can be found at `https://github.com/LamNgo1/mobo-osd`.

**PDBO [1].** We use the default hyperparameter settings from the paper, including the Hedge algorithm for selecting acquisition functions, the pool of acquisition functions EI, UCB, TS and Identity (i.e., the posterior mean function). We use the open-sourced implementation at `https://github.com/Alaleh/PDBO`.

**qEHVI [13].** We use the default hyperparameter settings from the paper. This includes the batch selection strategy that use sequential greedy approach to integrate over the unobserved outcomes. We use the open-sourced implementation at `https://github.com/pytorch/botorch`.

**DGEMO [33].** We use the default hyperparameter settings from the paper, including the number of buffer cells, max number of samples in each cell, buffer origin, graph-cut hyperparameters and solver NSGA-II hypeparameters. We use the open-sourced implementation at `https://github.com/yunshengtian/DGEMO`.

**USeMO [5].** We use the default hyperparameter settings from the paper. For the acquisition function, we use the EI [37] as it has the overall best performance and is widely compared in other works. We use the open-sourced implementation at `https://github.com/belakaria/USeMO`.

**qParEGO [31].** qParEGO is a novel extension from ParEGO [31] that is developed by Daulton et al. [13] to leverage batch setting. We use the settings as follows: augmented Tchebychev scalarization [39], EI acquisition function with gradient solver and sequential greedy batch selection strategy. We use the open-sourced implementation at `https://botorch.org/docs/tutorials`.

**JES [52].** We use the default hyperparameter settings from the paper. This includes the number of random initialization points for optimizing JES, and the NSGA-II hyperparameter settings for solving the Pareto front. We use the open-sourced implementation at `https://github.com/benmltu/JES`.

**NSGA-II [16].** We use NSGA-II implementation from `pymoo` [7]. We use the default settings as follows: population size of 100, binary tournament selection, simulated binary crossover (probability $p = 0.9$, exponential distribution parameter $\eta = 15$) and polynomial mutation (probability $p = 0.9$, exponential distribution parameter $\eta = 20$). We use the open-sourced implementation at `https://pymoo.org/algorithms/moo/nsga2.html`.

**NBI [11].** To the best of our knowledge, there is no available open-source implementation for NBI, therefore we recreated the implementation based on [11] as follows. At first, we sequentially optimize each objective $f_m$ to obtain the individual minima. Then the convex hull of individual minima (CHIM) can be computed via the convex combination weight vectors $\boldsymbol{\beta}$. We use the same Riesz s-Energy method [8] as in MOBO-OSD to generate exact $n_\beta = 20$ combination weight vectors, since the default point generation strategy cannot work with arbitrary $n_\beta$. See Appendix A.1 for details. Then, for each of $n_\beta$ NBI subproblems defined at each point on the CHIM, we solve the NBI subproblem to compute the intersections between the CHIM and the NBI normal search directions. Until the evaluation budget depletes, we solve $n_\beta$ NBI subproblems again (with new random initialization) to obtain new solutions. We aggregate all observations found, including when optimizing for the individual minima, when optimizing NBI subproblems and when evaluating the NBI subproblem constraints. Finally, from the observed dataset, we construct the approximate Pareto front for hypervolume comparison against other baselines.

As NBI requires evaluating objective functions for the constraints, the method is not sample-efficient. In fact, in our experiments, NBI often depletes the budget even when not having finished the first round of $n_\beta$ subproblems. We illustrate this statistics by presenting the number of function evaluations required to complete the individual minimum optimization and the first round of $n_\beta$ NBI subproblems given two types of solver, gradient-based SQLSP [35] and gradient-free COBYLA [43]). Details are shown in Table 3. We also present the performance of NBI given these two solvers in Fig. 9.

### A.8 Additional Results for Baseline Comparison

On top of the comparison results between MOBO-OSD and the baselines shown in Sec. 5.1, we provide extra results for other batch settings $b = \{8, 10\}$, which are shown in Figs 5 and 6, respectively. Note that qEVHI has limited iterations due to the prohibitively high computational cost (exceeding 12 hours runtime) and memory required (exceeding 64 GB) on problems with $M \geq 4$ objectives and batch size $b = \{8, 10\}$. MOBO-OSD and other baselines can finish the experiments within the given similar resources.

### A.9 Additional Results for Ablation Study

Figs 7 and 8 present the full results of the ablation study on $n_\beta$ parameter on all benchmark problems, on both sequential and batch settings, respectively. Overall, all variants have relatively similar performance, indicating the robustness of MOBO-OSD with respect to $n_\beta$ parameter.

### A.10 Computing Infrastructure

We run experiments on a computing server with a Dual CPU of type AMD EPYC 7662 (total of 128 Threads, 256 CPUs). Each experiment is allocated 8 CPUs and 64GB Memory. The server is installed with Ubuntu (20.04.3 LTS) Operating System.

### A.11 Computational Complexity

We provide the theoretical computational complexity analysis as follows. Let $N$ and $n$ denote the number of surrogate model training and testing points, respectively, and let $D$ and $M$ denote the number of dimensions and objectives, respectively. The computational complexity for the approximated CHIM and OSD formulation is $\mathcal{O}(NM)$ as these steps require iterating over $N$ data points with $M$-dimensional output. The GP training cost is well-known to scale cubically with the

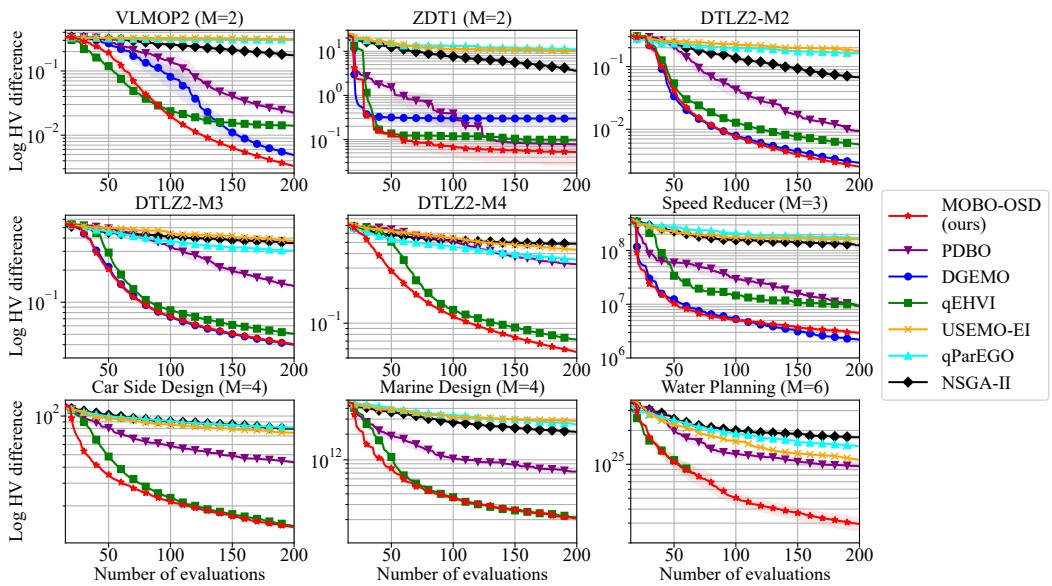

Figure 5: Comparison of MOBO-OSD against the SOTA baselines on 5 synthetic and 4 real-world benchmark problems in batch setting (batch size 8). Note that qEVHI has limited iterations due to the prohibitively high computational cost and memory required on problems with $M \geq 4$ objectives.

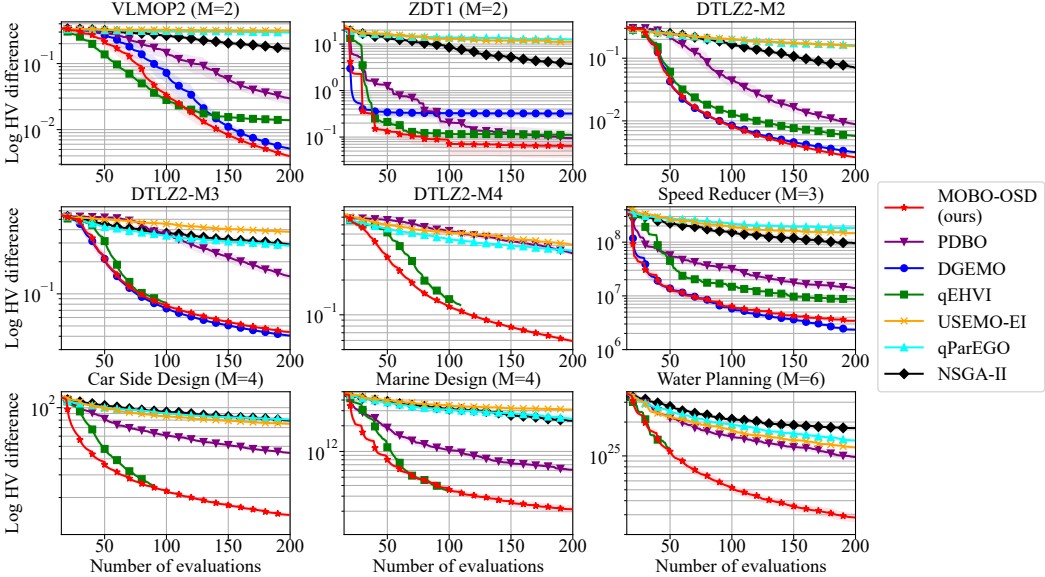

Figure 6: Comparison of MOBO-OSD against the SOTA baselines on 5 synthetic and 4 real-world benchmark problems in batch setting (batch size 10). Note that qEVHI has limited iterations due to the prohibitively high computational cost and memory required on problems with $M \geq 4$ objectives.

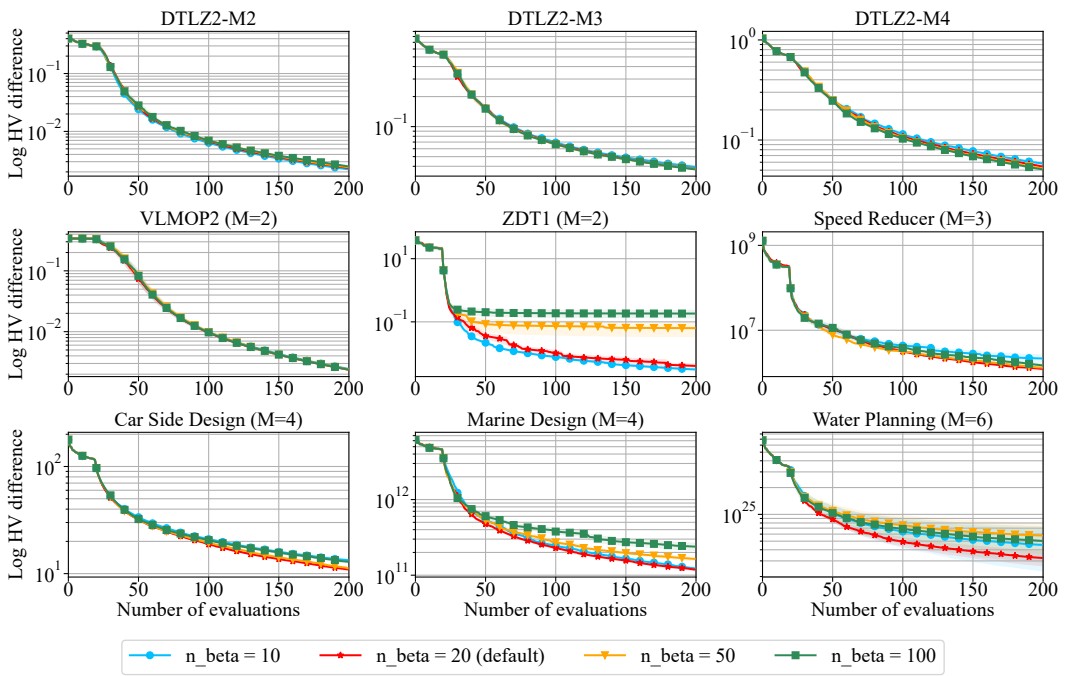

Figure 7: Ablation study on the effect of $n_\beta$ parameter (batch = 1). Overall, all variants have similar performance, indicating the robustness of MOBO-OSD with respect to $n_\beta$ parameter.

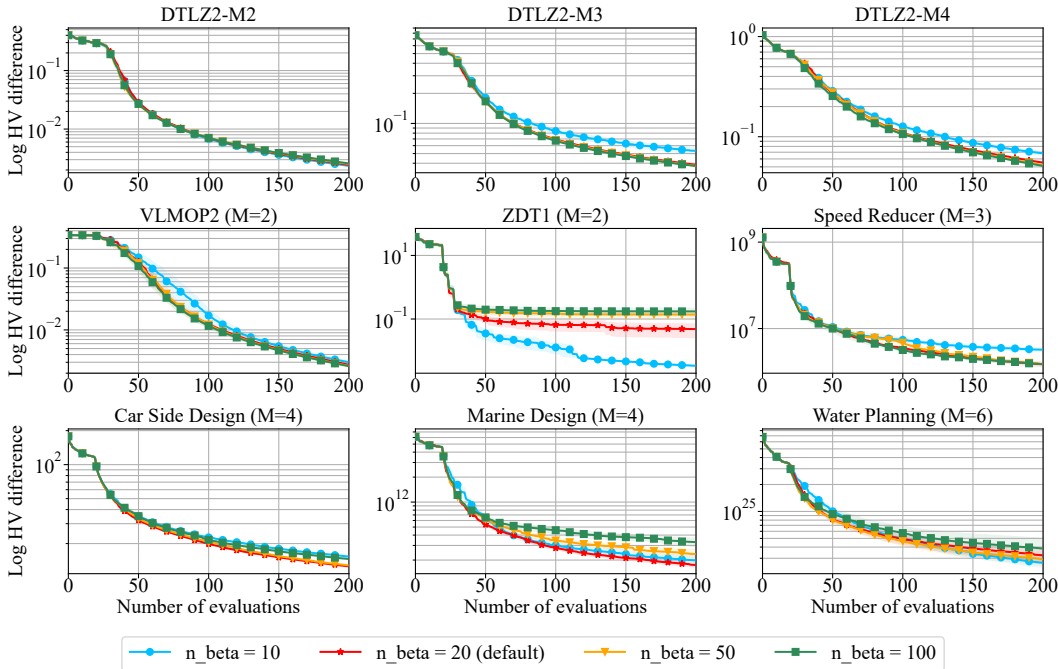

Figure 8: Ablation study on the effect of $n_\beta$ parameter (batch = 4). Overall, all variants have similar performance, indicating the robustness of MOBO-OSD with respect to $n_\beta$ parameter.

Table 3: Number of required function evaluations to complete the first round of $n_\beta$ NBI subproblems, including finding individual optima and solving $n_\beta$ NBI subproblems. Gradient-based solver cost significantly more budget due to gradient computation via finite difference method. Generally, NBI depletes all 200 evaluation budget before finishing the first round of optimization.

|  | Individual Objectives | Subproblems (Gradient-free solver) | Subproblems (Gradient-based solver) |
|---|---|---|---|
| VLMOP2 | $139.20 \pm 32.06$ | $259.20 \pm 32.06$ | $8049.70 \pm 1750.33$ |
| ZDT1-M2 | $24.00 \pm 0.00$ | $144.00 \pm 0.00$ | $720.20 \pm 29.50$ |
| DTLZ2-M2 | $30.00 \pm 9.30$ | $150.00 \pm 9.30$ | $898.20 \pm 22.52$ |
| DTLZ2-M3 | $36.60 \pm 1.80$ | $156.60 \pm 1.80$ | $1118.90 \pm 213.97$ |
| DTLZ2-M4 | $54.00 \pm 8.90$ | $174.00 \pm 8.90$ | $2050.80 \pm 1176.62$ |
| Speed Reducer | $72.00 \pm 25.04$ | $232.00 \pm 25.04$ | $7680.70 \pm 1023.34$ |
| Car Side Design | $109.60 \pm 10.15$ | $269.60 \pm 10.15$ | $5141.30 \pm 1105.41$ |
| Marine Design | $139.30 \pm 22.45$ | $279.30 \pm 22.45$ | $7859.30 \pm 1032.65$ |
| Water Planning | $44.00 \pm 0.00$ | $124.00 \pm 0.00$ | $124.00 \pm 0.00$ |

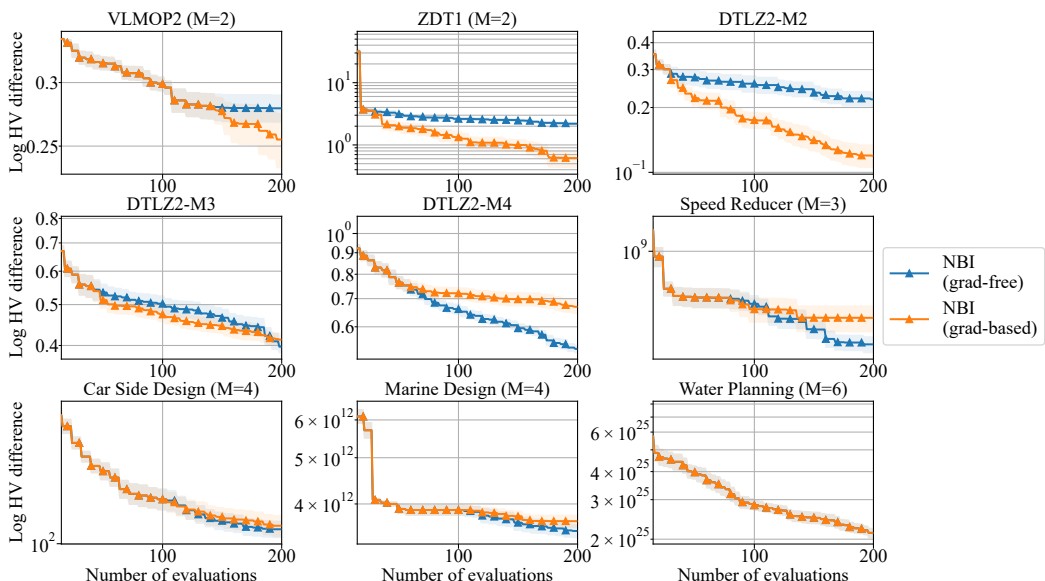

Figure 9: Performance of NBI when using different solvers for the NBI subproblems. Overall, no variant is consistently better than the other.

number of training samples, resulting in a computational complexity of $\mathcal{O}(N^3)$. The computational complexity for solving each MOBO-OSD subproblem is $\mathcal{O}(D^3 + N^2 n)$, which comprises of two dominated components: the SLSQP solver with cost of $\mathcal{O}(D^3)$, and the approximated cost of $\mathcal{O}(N^2 n)$ associated with evaluating the mean vector, standard deviation vector, and their corresponding gradients. For the Pareto Front Estimation step, following Schulz et al. [46] and Konakovic Lukovic et al. [33], the computational cost is $\mathcal{O}(D^3 + N n)$. Finally, for the batch Hypervolume Improvement acquisition function, following Daulton et al. [13], the computational complexity is $\mathcal{O}(MK(2^b - 1))$, where $K$ is the number of disjoint partitions for box decompositions and $b$ is the batch size.

Additionally, we provide the empirical runtime for the proposed method and all baselines in Tables 4, 5, 6 and 7 for batch size $b = \{1, 4, 8, 10\}$, respectively. Note that qEHVI is significantly more computationally (in terms of runtime and required memory) than other baselines and MOBO-OSD, especially when $b = \{8, 10\}$, hence can only have partial results on problems with $M \geq 4$ objectives. All NSGA-II runs cost less than 0.01s, so we exclude the method from the results. Moreover, symbol "-" indicates the method is not applicable for the corresponding baseline, according to the authors' implementation.

Table 4: Runtime comparison (seconds per iteration) for *batch size 1*. Overall, our proposed method, MOBO-OSD, have affordable time complexity. JES is significantly more computational expensive even in sequential setting. Symbol "-" indicates the method is not applicable for the corresponding benchmark problems.

| | MOBO-OSD (Ours) | PDBO | qEHVI | JES | DGEMO | USeMO (EI) | qParEGO |
|---|---|---|---|---|---|---|---|
| VLMOP2 | 16.18 | 32.02 | 2.14 | 67.30 | 9.46 | 1.89 | 0.60 |
| ZDT1-M2 | 7.50 | 22.47 | 1.89 | 40.67 | 11.84 | 2.28 | 0.50 |
| DTLZ2-M2 | 8.34 | 23.74 | 2.95 | 46.99 | 8.74 | 1.91 | 0.50 |
| DTLZ2-M3 | 11.72 | 15.38 | 10.32 | 53.69 | 10.63 | 2.75 | 0.68 |
| DTLZ2-M4 | 19.85 | 14.66 | 55.27 | 126.97 | - | 3.75 | 0.88 |
| Speed Reducer | 17.18 | 13.38 | 6.91 | 115.07 | 16.31 | 2.67 | 0.71 |
| Car Side Design | 24.84 | 15.65 | 16.31 | 105.03 | - | 4.00 | 0.92 |
| Marine Design | 22.72 | 17.03 | 17.17 | 91.56 | - | 3.50 | 1.81 |
| Water Planning | 30.16 | 73.66 | 150.93 | 846.36 | - | 5.34 | 1.72 |

Table 5: Runtime comparison (seconds per iteration) for *batch size 4*. Overall, our proposed method, MOBO-OSD, have affordable time complexity. qEHVI is significantly more computational expensive with increasing batch size. Symbol "-" indicates the method is not applicable for the corresponding benchmark problems.

| | MOBO-OSD (Ours) | PDBO | qEHVI | DGEMO | USeMO (EI) | qParEGO |
|---|---|---|---|---|---|---|
| VLMOP2 | 4.34 | 8.23 | 2.20 | 2.33 | 0.69 | 0.45 |
| ZDT1-M2 | 2.31 | 5.86 | 1.63 | 2.81 | 0.65 | 0.39 |
| DTLZ2-M2 | 2.51 | 7.23 | 2.60 | 2.13 | 0.53 | 0.39 |
| DTLZ2-M3 | 3.80 | 4.95 | 8.82 | 2.86 | 0.78 | 0.52 |
| DTLZ2-M4 | 10.58 | 4.45 | 31.73 | - | 1.14 | 0.65 |
| Speed Reducer | 4.65 | 3.84 | 14.06 | 3.96 | 0.72 | 0.53 |
| Car Side Design | 13.90 | 5.11 | 71.46 | - | 1.23 | 0.66 |
| Marine Design | 10.03 | 5.38 | 57.27 | - | 1.05 | 1.83 |
| Water Planning | 24.25 | 21.47 | 272.16 | - | 2.06 | 1.11 |

Table 6: Runtime comparison (seconds per iteration) for *batch size 8*. Overall, our proposed method, MOBO-OSD, have affordable time complexity. qEHVI is significantly more computational expensive with increasing batch size. Symbol "-" indicates the method is not applicable for the corresponding benchmark problems.

| | MOBO-OSD (Ours) | PDBO | qEHVI | DGEMO | USeMO (EI) | qParEGO |
|---|---|---|---|---|---|---|
| VLMOP2 | 1.69 | 3.98 | 7.25 | 1.07 | 0.28 | 0.35 |
| ZDT1-M2 | 1.11 | 3.05 | 4.47 | 1.35 | 0.42 | 0.34 |
| DTLZ2-M2 | 1.14 | 3.56 | 10.00 | 1.03 | 0.33 | 0.34 |
| DTLZ2-M3 | 1.71 | 2.22 | 100.07 | 1.43 | 0.55 | 0.46 |
| DTLZ2-M4 | 2.42 | 2.33 | 449.26 | - | 0.66 | 0.56 |
| Speed Reducer | 2.42 | 2.36 | 66.35 | 2.16 | 0.36 | 0.46 |
| Car Side Design | 2.50 | 3.22 | 554.83 | - | 0.78 | 0.55 |
| Marine Design | 3.13 | 2.97 | 374.85 | - | 0.50 | 1.64 |
| Water Planning | 11.64 | 11.97 | 651.57 | - | 0.97 | 0.79 |

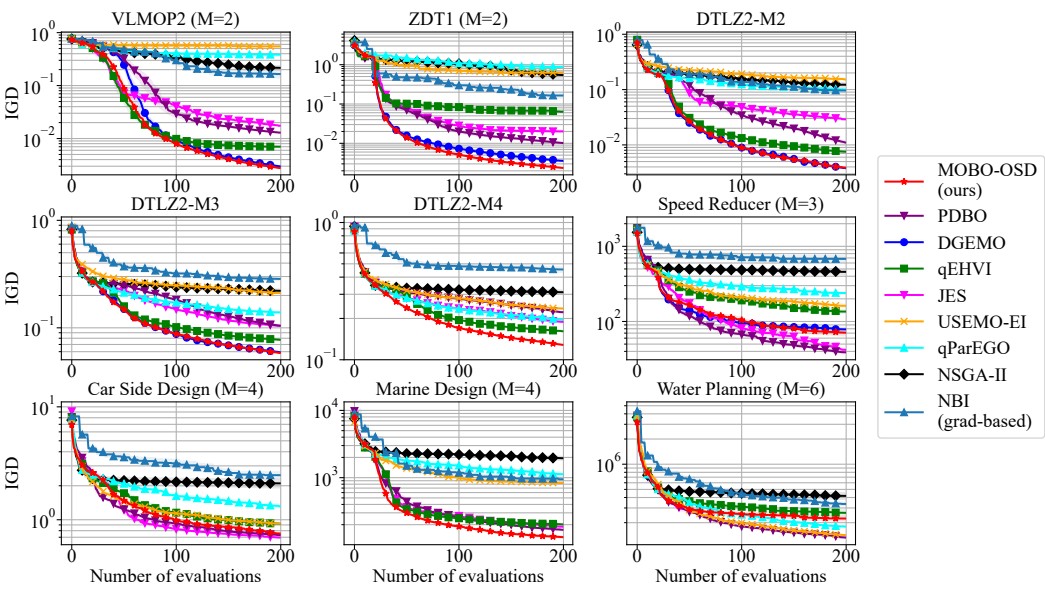

Figure 10: Comparison using *IGD indicator* of MOBO-OSD against the baselines on 5 synthetic and 4 real-world benchmark problems with batch size 1.

Table 7: Runtime comparison (seconds per iteration) for *batch size 10*. Overall, our proposed method, MOBO-OSD, have affordable time complexity. qEHVI is significantly more computational expensive with increasing batch size. Symbol "-" indicates the method is not applicable for the corresponding benchmark problems.

| | MOBO-OSD (Ours) | PDBO | qEHVI | DGEMO | USeMO (EI) | qParEGO |
|---|---|---|---|---|---|---|
| VLMOP2 | 1.71 | 3.29 | 45.84 | 0.90 | 0.38 | 0.41 |
| ZDT1-M2 | 1.13 | 2.57 | 18.72 | 1.11 | 0.67 | 0.41 |
| DTLZ2-M2 | 1.15 | 2.92 | 57.72 | 0.82 | 0.45 | 0.40 |
| DTLZ2-M3 | 1.78 | 1.89 | 229.09 | 1.25 | 0.46 | 0.55 |
| DTLZ2-M4 | 2.73 | 2.07 | 446.87 | - | 1.00 | 0.70 |
| Speed Reducer | 2.49 | 1.89 | 158.63 | 1.67 | 0.42 | 0.54 |
| Car Side Design | 2.77 | 2.47 | 498.51 | - | 0.87 | 0.67 |
| Marine Design | 3.48 | 2.35 | 468.98 | - | 0.61 | 2.42 |
| Water Planning | 21.78 | 11.20 | 710.14 | - | 1.19 | 0.94 |

## A.12   Other Comparison Metrics

We compare the performance of MOBO-OSD and the baselines using other comparison metrics, including IGD [10], IGD+ [25] and $\varepsilon$-indicator [60]. Figs 10, 11, 12 and 13 show the IGD metrics. Figs 14, 15, 16 and 17 show the IGD+ metrics. Figs 18, 19, 20 and 21 show the $\varepsilon$-indicator. Overall, MOBO-OSD shows competitive performance regardless of performance metrics.

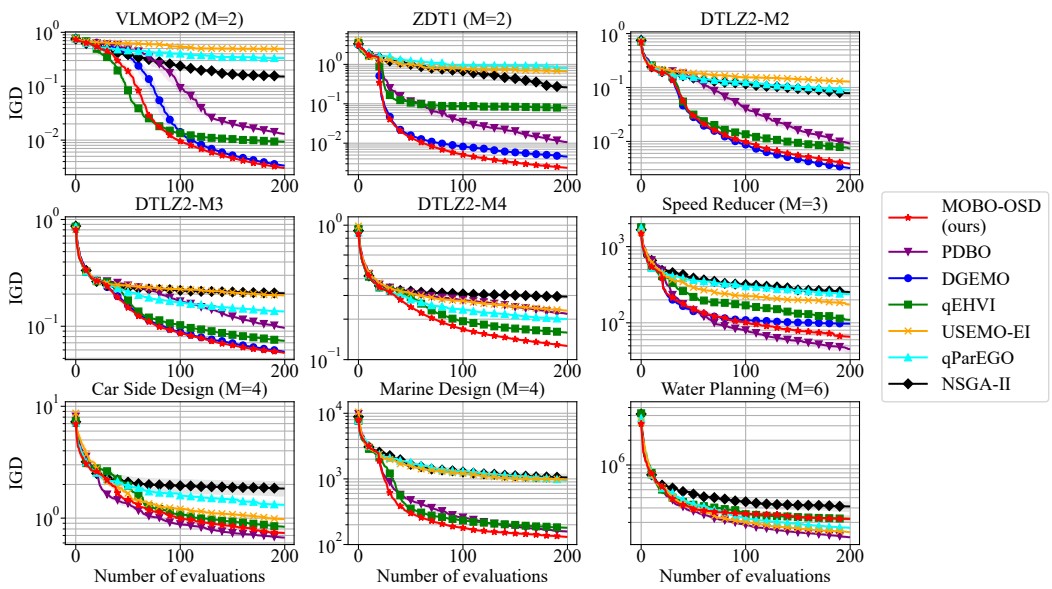

Figure 11: Comparison using *IGD indicator* of MOBO-OSD against the baselines on 5 synthetic and 4 real-world benchmark problems with batch size 4.

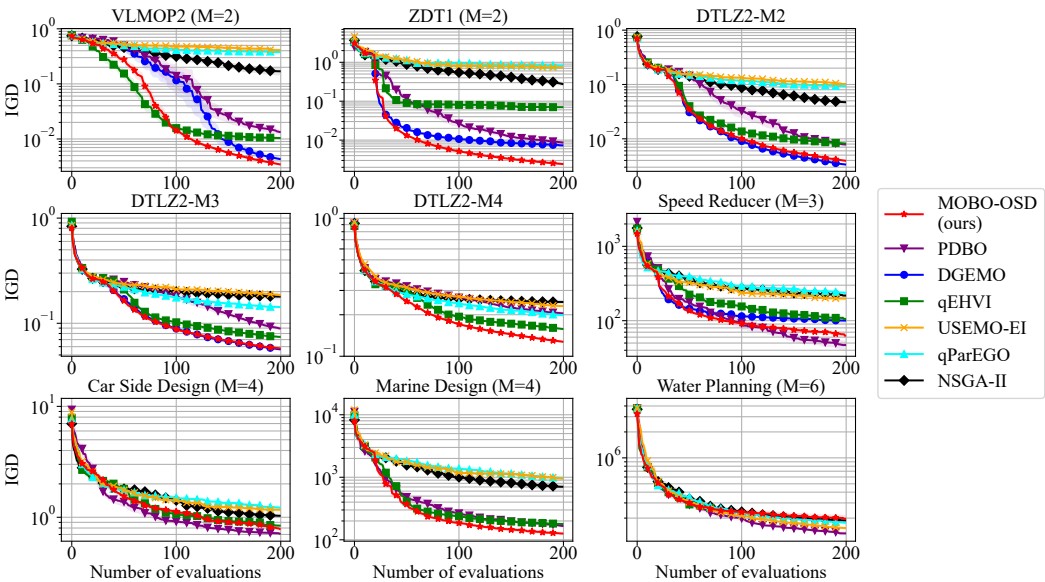

Figure 12: Comparison using *IGD indicator* of MOBO-OSD against the baselines on 5 synthetic and 4 real-world benchmark problems with batch size 8.

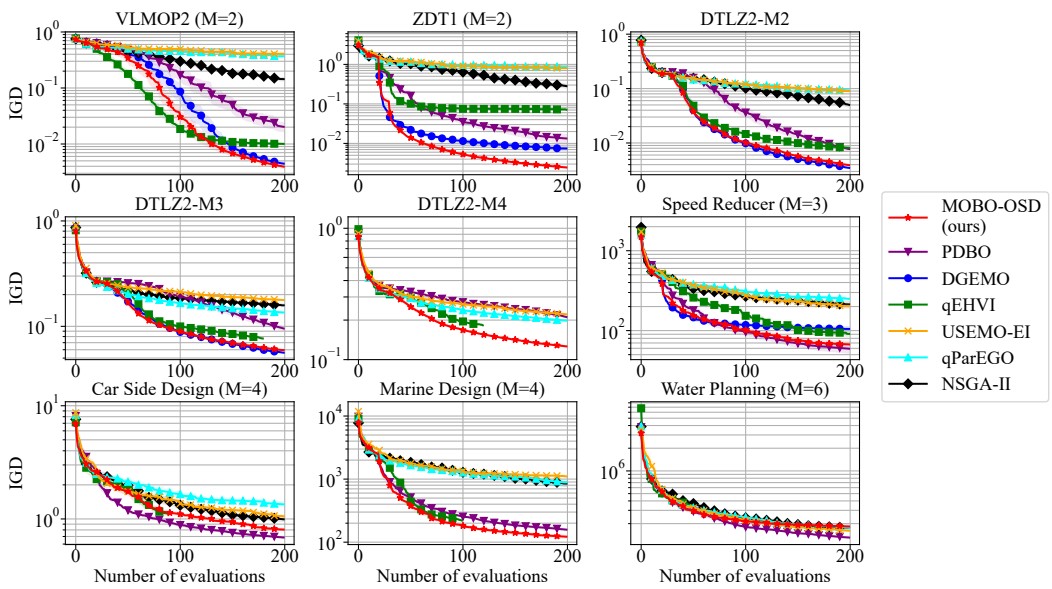

Figure 13: Comparison using *IGD indicator* of MOBO-OSD against the baselines on 5 synthetic and 4 real-world benchmark problems with batch size 10.

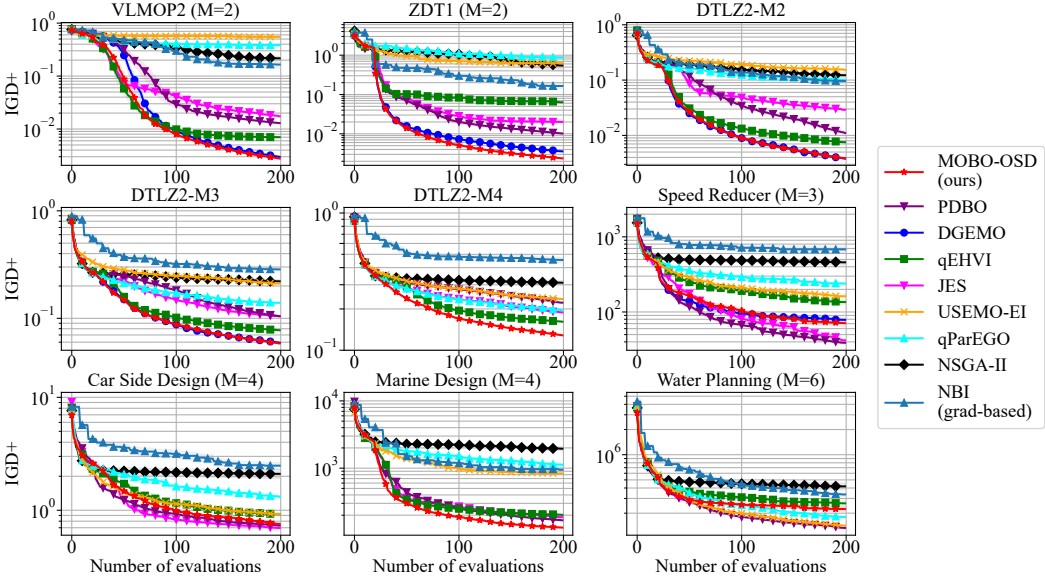

Figure 14: Comparison using *IGD+ indicator* of MOBO-OSD against the baselines on 5 synthetic and 4 real-world benchmark problems with batch size 1.

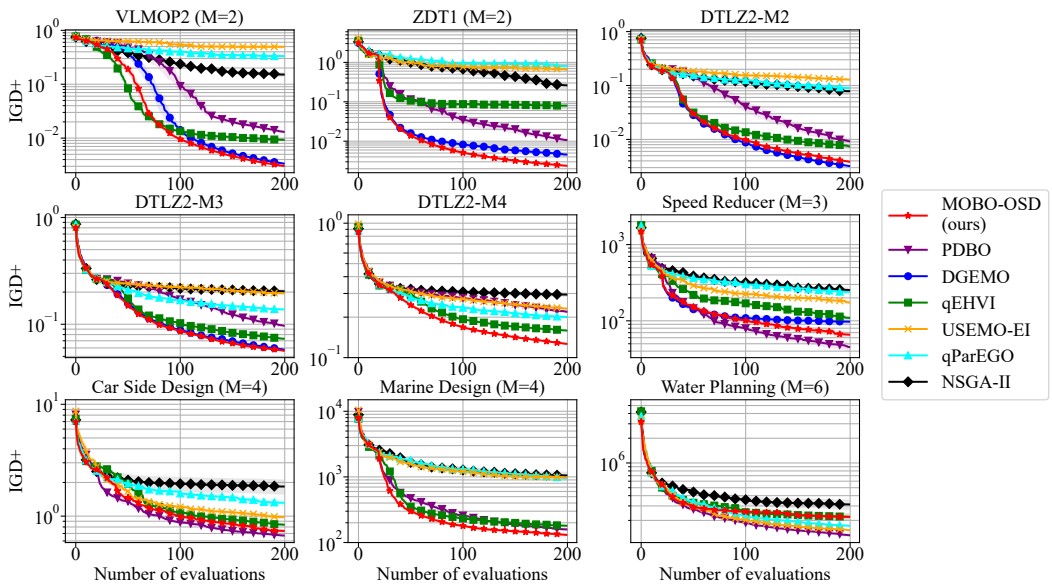

Figure 15: Comparison using *IGD+ indicator* of MOBO-OSD against the baselines on 5 synthetic and 4 real-world benchmark problems with batch size 4.

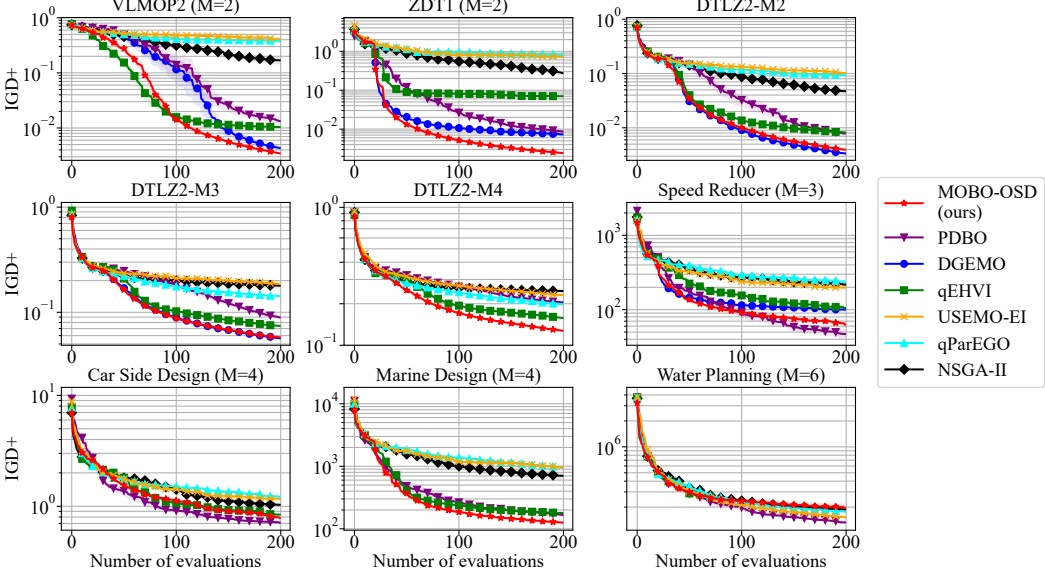

Figure 16: Comparison using *IGD+ indicator* of MOBO-OSD against the baselines on 5 synthetic and 4 real-world benchmark problems with batch size 8.

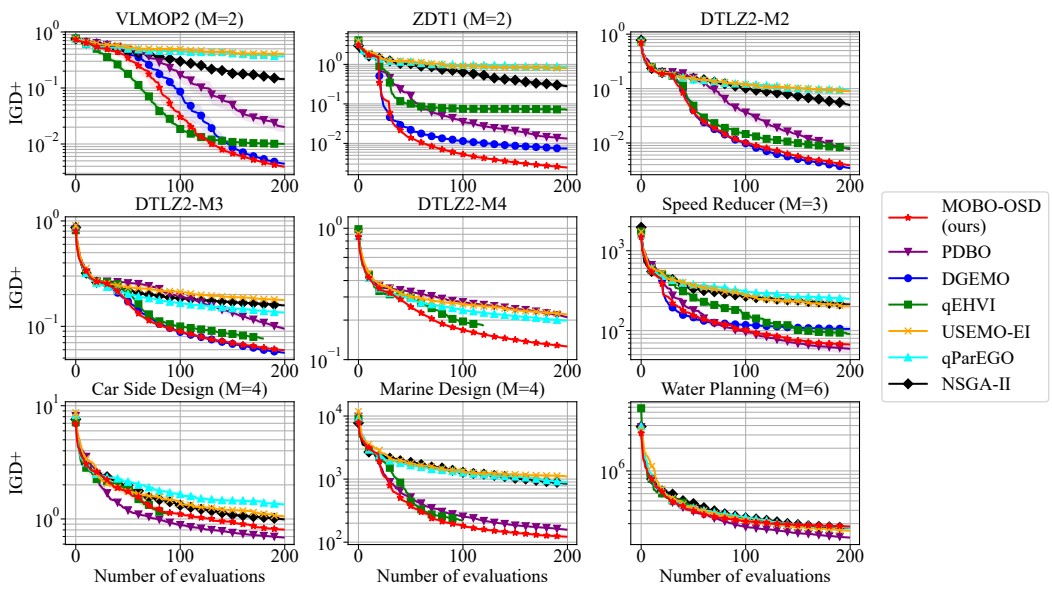

Figure 17: Comparison using *IGD+ indicator* of MOBO-OSD against the baselines on 5 synthetic and 4 real-world benchmark problems with batch size 10.

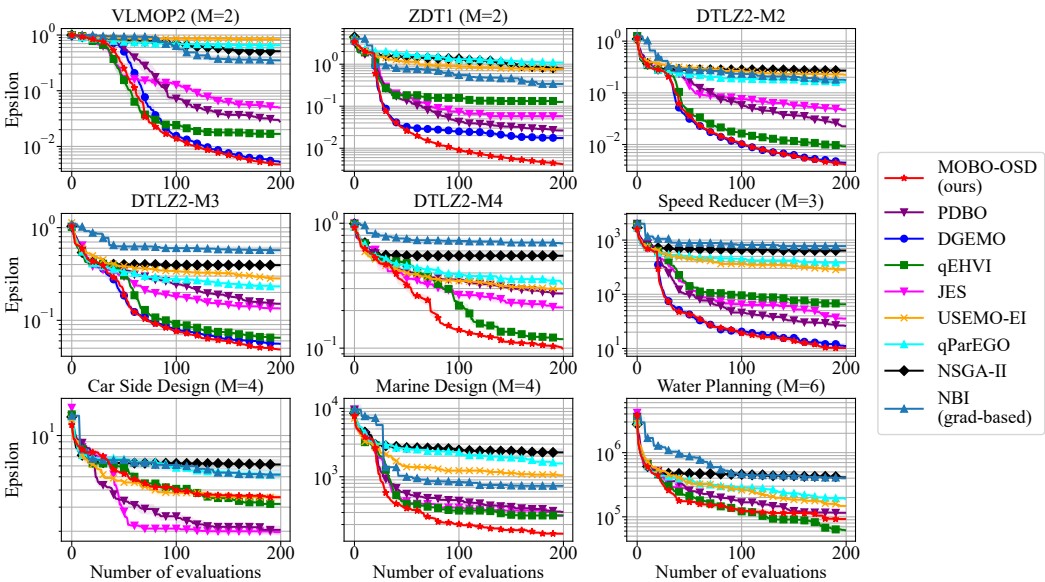

Figure 18: Comparison using *ε-indicator* of MOBO-OSD against the baselines on 5 synthetic and 4 real-world benchmark problems with batch size 1.

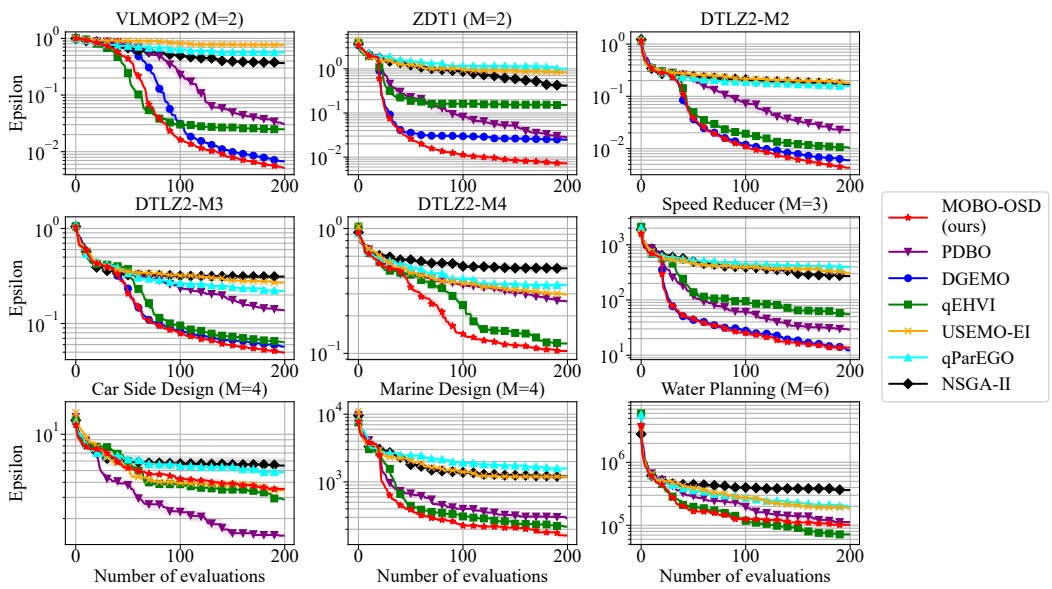

Figure 19: Comparison using $\varepsilon$-*indicator* of MOBO-OSD against the baselines on 5 synthetic and 4 real-world benchmark problems with batch size 4.

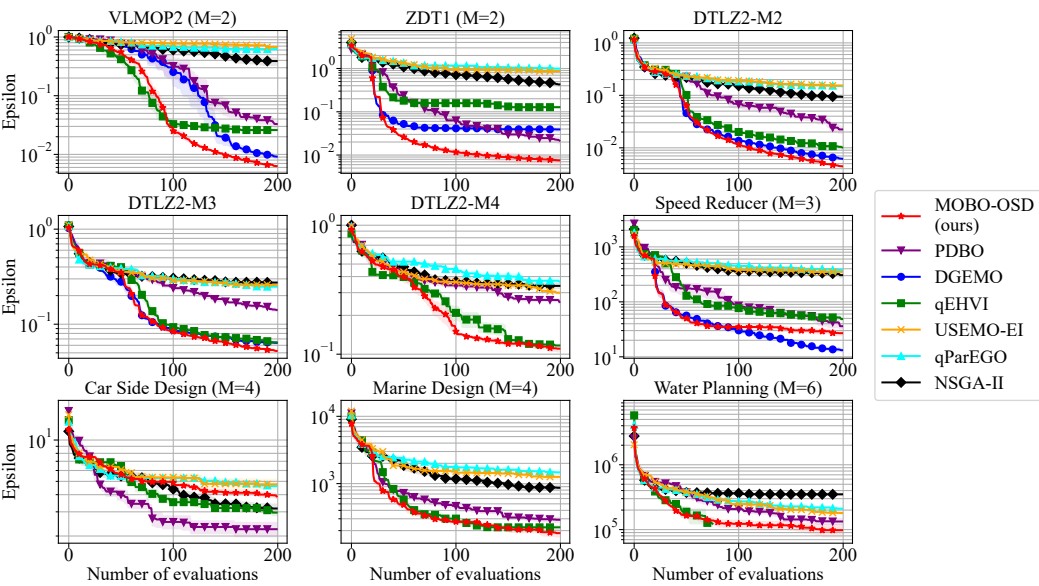

Figure 20: Comparison using $\varepsilon$-*indicator* of MOBO-OSD against the baselines on 5 synthetic and 4 real-world benchmark problems with batch size 8.

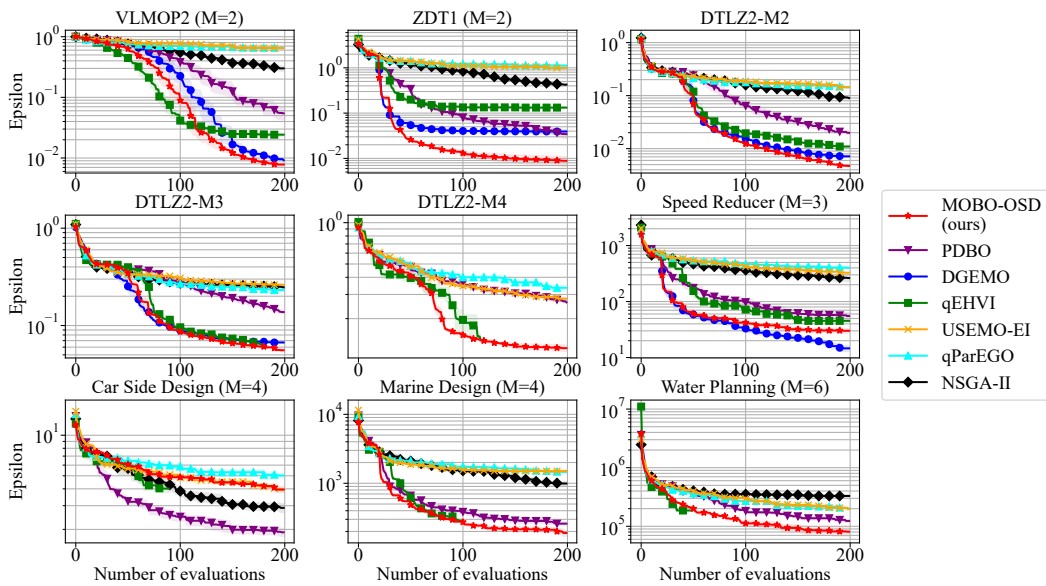

Figure 21: Comparison using $\varepsilon$-*indicator* of MOBO-OSD against the baselines on 5 synthetic and 4 real-world benchmark problems with batch size 10.

