# OpenReview forum: "MOBO-OSD: Batch Multi-Objective Bayesian Optimization via Orthogonal Search Directions"
_NeurIPS.cc/2025/Conference — NeurIPS 2025 poster_

### Official Review · Reviewer_RbX4 · 2025-06-03

**Clarity:** 3
**Significance:** 3
**Originality:** 3
**Rating:** 5
**Confidence:** 3

**Summary:**

The authors propose “Batch Multi-Objective Bayesian Optimization via Orthogonal Search Directions” (MOBO-OSD), a novel Bayesian optimization method for expensive, multi-objective black-box optimization problems. MOBO-OSD solves multiple constrained optimization problems along orthogonal search directions (OSDs) in order to obtain a final diverse set of Pareto optimal solutions. These multiple constrained problems are called MOBO-OSD subproblems. Three key strategies are used by MOBO-OSD to obtain good performance. First, to maintain a broad search region, MOBO-OSD approximates the Convex Hull of Individual Objective Minima (CHIM) using the best (ideal) and worst (nadir) observed points. Second, MOBO-OSD generates a well-distributed set of OSDs using the Riesz s-Energy method. Third, to further enhance density and diversity of solutions without requiring an excessive number of subproblems, MOBO-OSD uses Pareto Front Estimation (PFE) for local exploration around candidate solutions. The authors also extend MOBO-OSD to the batch optimization setting and use the Kriging Believer method of considering distinct exploration regions in order to enhance diversity in the selected batch. The authors provide extensive experiments on various synthetic and real-world benchmark tasks with between 2 and 6 objectives. Results demonstrate MOBO-OSD consistently outperforms SOTA baseline methods in both the sequential and batch optimization settings.

**Questions:**

Question 1:
MOBO-OSD approximates the CHIM with an ideal point and a nadir point to define boundries. Can the authors elaborate on the choice of using the nadir point (the worst observed value for each objective)? How does this specific choice encourage a "broader search region"?


Question 2:
The authors provide an ablation study showing that MOBO-OSD is robust to the number of OSDs, largely due to the PFE component. What are the practical trade-offs a practitioner should consider when applying MOBO-OSD in real-world problem settings. In particular, when deciding between using PFE with fewer OSDs, versus using more OSDs possibly without PFE, how might this impact computational cost and diversity/performance of the final solutions obtained?


Question 3:
The authors cite their focus on settings with noiseless observations as a limitation of MOBO-OSD. What do you foresee as the primary challenges in adapting the proposed MOBO-OSD to handle noisy observations effectively?

**Ethical Concerns:**

["NO or VERY MINOR ethics concerns only"]

**Final Justification:**

In their rebuttal, the authors answer all of my questions sufficiently. I maintain that this work should be accepted for all of the reasons stated in my initial review. Additionally, the other reviews and the author's rebuttals to them served to confirm my initial assessment, it seems that all the reviewers are in agreement that this paper should be accepted. I therefore maintain my score of 5.

**Limitations:**

yes

**Quality:**

3

**Strengths And Weaknesses:**

Quality:
The work is high quality. Figures and text are well polished. Ablation studies are extensive and provide useful insight.

Clarity:
The work is very well written and the author's method and motivation are clearly explained.

Significance:
This work is significant largely due to the strength of the empirical results. The proposed MOBO-OSD clearly outperforms SOTA baselines in the literature across tasks. This is further strengthened by the inclusion of the 4 real-world benchmark problems. Since multi-objective black-box optimization is particularly relevant and applicable to many areas of the broader research community, this work will have significant impact. The primary weakness of the method is that it cannot be readily applied to settings with noisy observations and this is the case in many real-world settings. However, the authors acknowledge this limitation and I think that this is a very minor weakness of the work overall.

Originality:
This work is clearly novel since it prevents a novel strategy to handle the problem of finding a diverse set of pareto-optimal solutions for multi-objective black-box optimization problems.

---

> ### Author Rebuttal · Authors · 2025-07-31
>
> We thank the reviewer for their insightful review. We address the questions and comments as follows:
>
> # Questions
>
> > Question 1: MOBO-OSD approximates the CHIM with an ideal point and a nadir point to define boundaries. Can the authors elaborate on the choice of using the nadir point (the worst observed value for each objective)? How does this specific choice encourage a "broader search region"?
>
> As mentioned in Section 4.1, our proposed approximated Convex Hull of Individual Minima (CHIM) - constructed from the proposed boundary points defined from ideal and nadir points - can generate a larger search region compared to the alternative CHIM constructed from the individual minima of the current observed dataset. Note that we only consider the alternative CHIM as the true CHIM requires the unknown true individual minima. By definition, both individual minima and boundary points contain one component of the ideal point (as illustrated by the green and purple stars in the inset of Section 4.1). Therefore, by setting the remaining components of each boundary point to the worst (largest) value possible in the dataset, i.e., the nadir points, our proposed boundary points, and consequently the approximated CHIM, can define a larger search region than that generated using the individual minima and the alternative CHIM. We will update this explanation in the revised version of the paper.
>
> > Question 2: The authors provide an ablation study showing that MOBO-OSD is robust to the number of OSDs, largely due to the PFE component. What are the practical trade-offs a practitioner should consider when applying MOBO-OSD in real-world problem settings. In particular, when deciding between using PFE with fewer OSDs, versus using more OSDs possibly without PFE, how might this impact computational cost and diversity/performance of the final solutions obtained?
>
> A practical trade-off worth considering is that increasing the number of OSDs results in better performance but also incurs higher computational cost, hence *we recommend using PFE with fewer OSDs* as the default setting for all real-world problems. This is because, as shown in the two ablation studies in Section 5.1.2, incorporating PFE enables MOBO-OSD to maintain strong performance without having to solve an excessive number OSD subproblems.
>
> > Question 3: The authors cite their focus on settings with noiseless observations as a limitation of MOBO-OSD. What do you foresee as the primary challenges in adapting the proposed MOBO-OSD to handle noisy observations effectively?
>
> As mentioned in in the Limitations section of the Conclusion, the primary challenge lies in the acquisition function (AF). This is because the Hypervolume Improvement AF depends on the posterior mean, which is sensitive to noise. Another challenge lies in the MOBO-OSB subproblems (Eq. (2)), as solving Eq. (2) requires estimating the posterior mean, standard deviation, and their gradients, all of which are sensitive to noise. Additionally, the current integrated PFE might pose another limitation, as the algorithm in [42] was originally designed for true objective function values (not surrogate model estimates). Therefore, applying it directly to noisy surrogate predictions may compromise the quality of the estimated Pareto front.

---

> > ### Comment · Reviewer_RbX4 · 2025-08-04
> >
> > I would like to thank the authors for taking time to answer each of my questions. I maintain that this work should be accepted to NeurIPS.

---

> > > ### Author Response · Authors · 2025-08-06
> > >
> > > We would like to thank the reviewer for acknowledging our response, and we are glad that our clarification addressed your concerns.

---

### Official Review · Reviewer_xpJo · 2025-06-29

**Clarity:** 3
**Significance:** 2
**Originality:** 3
**Rating:** 5
**Confidence:** 3

**Summary:**

Their method is based on NBI, which computes the CHIM (convex hull of the individual minima) and works well when there is plenty of budget. Since MOBO problems have limited budget, they propose a technique to efficiently approximate the CHIM. Starting from this approximated CHIM, they generate orthogonal search directions (OSDs) evenly distributed, defined as constrained single-objective subproblems in which they look for the intersection points (candidates) with the Pareto front generated by the GPs. After solving these subproblems, they employ the Pareto Front Estimation (PFE) technique to increase the number of candidates (instead of solving more subproblems). Then, for each candidate they calculate its HVI, and if batch > 1 they use Kriging Believer to choose the rest of the batch candidates one by one, ensuring diversity among candidates. Finally, they claim that their method performs as well as or better than the state-of-the-art and that it can scale to any number of objectives in both sequential and batch evaluations.

**Questions:**

Have you considered using IGD [4], IGD^+ [5], and the ε-indicator [6] alongside HVI for batch selection? HVI maximizes dominated volume, IGD/IGD^+ ensure uniform coverage, and the ε-indicator guarantees worst-case bounds. What are your thoughts?

[4] Coello, C. A. C., & Lamont, G. B. (2004). Applications of multi-objective evolutionary algorithms (Vol. 1). World Scientific.

[5] Ishibuchi, H., Masuda, H., Tanigaki, Y., & Nojima, Y. (2015). Modified distance calculation in generational distance and inverted generational distance. In Evolutionary Multi-Criterion Optimization: 8th International Conference, EMO 2015, Guimarães, Portugal, March 29--April 1, 2015. Proceedings, Part II 8 (pp. 110-125). Springer International Publishing.

[6] Zitzler, E., Thiele, L., Laumanns, M., Fonseca, C. M., & Da Fonseca, V. G. (2003). Performance assessment of multiobjective optimizers: An analysis and review. IEEE Transactions on evolutionary computation, 7(2), 117-132.

--

These sentences sound strange:

 - Many existing approaches rely on scalarization techniques [5, 8, 36], which often fail to capture a diverse Pareto front, leading to suboptimal **performance in performance** such as hypervolume.

- In order to generate a **well-distributed** set of OSDs, it is essential to construct a **well-distributed** set of U(β) points. This is equivalent to generate a **well-distributed** set of {β} over a M -dimensional unit simplex, as U(β), by definition, is the linear transformation of β from a unit simplex to the approximated CHIM.

**Ethical Concerns:**

["NO or VERY MINOR ethics concerns only"]

**Final Justification:**

After reading the authors' responses to me and the other reviewers, I've decided to raise my score.

**Limitations:**

I have described the limitations I found in the Weaknesses section.

**Paper Formatting Concerns:**

I think there are no formatting problems.

**Quality:**

3

**Strengths And Weaknesses:**

Strengths:

 - Their explanatory figures help to understand their method.

 - They have run several benchmarks from the literature to test their method in both sequential and batch scenarios.

 - Their method outperforms all other methods on all the benchmarks they present.

Weaknesses:

 - Their method only handles problems without noise.

 - They mention JES [1] and PFES [2] in the "Related Work" section but they do not include them in the experiments. While these methods can be somewhat costly, I think it is important to include them for M <= 4 in the sequential, non-batch, problems.

- The greedy selection for the batch of points to evaluate is not the best option. Although the Hypervolume Subset Selection Problem is #P-hard, there are better alternatives [3].

[1] Ben Tu, Axel Gandy, Nikolas Kantas, and Behrang Shafei. Joint entropy search for multi- objective bayesian optimization. Advances in Neural Information Processing Systems, 35: 9922–9938, 2022.

[2] Shinya Suzuki, Shion Takeno, Tomoyuki Tamura, Kazuki Shitara, and Masayuki Karasuyama. Multi-objective bayesian optimization using pareto-frontier entropy. In International conference on machine learning, pages 9279–9288. PMLR, 2020.

[3] Guerreiro, Andreia P., Fonseca, Carlos M., & Paquete, Luís (2021). The Hypervolume Indicator: Computational Problems and Algorithms. ACM Computing Surveys (CSUR), 54(6), 1–42.

---

> ### Author Rebuttal · Authors · 2025-07-31
>
> We thank the reviewer for their insightful review. We address the questions and comments as follows:
>
> # Questions
>
> > Have you considered using IGD, IGD+, and the $\varepsilon$-indicator alongside HVI for batch selection? HVI maximizes dominated volume, IGD/IGD+ ensure uniform coverage, and the $\varepsilon$-indicator guarantees worst-case bounds. What are your thoughts?
>
> The Hypervolume Improvement (HVI) metric is widely recognized as the most commonly used performance metric in general multi-objective optimization (MOO) and, in particular, multi-objective Bayesian optimization (MOBO) [29, 37, 5, 1, 22, 48, 12, 31]. Nonetheless, we agree that presenting additional metrics can provide further insights into our proposed method. We have now conducted comparisons using IGD, IGD+ and $\varepsilon$-indicator metrics across all baselines, all benchmark problems, and batch settings. The IGD and IGD+ results show that MOBO-OSD converges rapidly towards 0, indicating the uniform coverage of the discovered Pareto fronts. The $\varepsilon$-indicator results exhibit convergence toward 0, reflecting strong worst-case guarantees. Compared to the baselines, overall, MOBO-OSD still outperforms them under these new metrics.  We will include these additional metrics in the revised version of the paper.
>
> > These sentences sound strange…
>
> Thank you for the comment. We will fix these sentences in the revised version of the paper.
>
> # Weakness
>
> > Their method only handles problems without noise.
>
> We agree with this limitation, which has already been acknowledged in the Conclusion section of our manuscript. It is important to note that this limitation is not unique to our method but is also commonly encountered in existing MOBO approaches, including recent state-of-the-art baselines such as qEHVI [12] and DGEMO [31]. Reviewer RbX4 also agrees that this is a very minor weakness. Nonetheless, addressing noisy settings remains an important challenge and is a promising direction for our future work.
>
> > They mention JES [1] and PFES [2] in the "Related Work" section but they do not include them in the experiments. While these methods can be somewhat costly, I think it is important to include them for M <= 4 in the sequential, non-batch, problems.
>
> The main reason we did not include PFES and JES is that they are information-theoretic (IT)-based acquisition functions (AFs), whereas our proposed MOBO-OSD uses a hypervolume-based (HV-based) AF. As shown in Figures 2, 3, 8 and 9, we have already empirically demonstrated that MOBO-OSB outperforms other state-of-the-art HV-based baselines such as qEHVI and DGEMO. Nonetheless, we appreciate the reviewer’s suggestion and would like to note that we have now evaluated JES (the most recent IT-based AF) on all nine benchmark problems under sequential settings (batch size 1). The results are summarized in the following table, which reports the mean and standard error of the log HV difference between JES and MOBO-OSD at the end of the optimization run (200 iterations). Overall, *MOBO-OSD significantly outperforms JES on all nine problems*.
>
> | | MOBO-OSD            | JES                 |
> |:----------------|:--------------------|:--------------------|
> | DTLZ2-M2        | 2.34e-03 ± 4.42e-05 | 3.32e-02 ± 6.46e-04 |
> | DTLZ2-M3        | 3.78e-02 ± 2.33e-04 | 1.31e-01 ± 1.70e-03 |
> | DTLZ2-M4        | 5.45e-02 ± 8.00e-04 | 2.14e-01 ± 3.36e-03 |
> | VLMOP2          | 2.39e-03 ± 3.59e-05 | 3.11e-02 ± 8.67e-04 |
> | ZDT1            | 4.03e-03 ± 5.55e-04 | 3.38e-02 ± 2.41e-03 |
> | Speed Reducer   | 1.24e+06 ± 7.70e+04 | 8.85e+06 ± 5.75e+05 |
> | Car Side Design | 1.10e+01 ± 2.62e-01 | 4.44e+01 ± 5.89e-01 |
> | Marine Design   | 1.19e+11 ± 3.42e+09 | 5.80e+11 ± 2.36e+10 |
> | Water Planning  | 3.24e+24 ± 5.82e+23 | 1.49e+25 ± 8.10e+23 (*) |
>
> Additionally, as the reviewer noted, JES suffers from high computational cost and is unable to complete the Water Planning problem ($M=6$). The asterisk in the table denotes results based on iteration data available before termination due to the time limit. Specifically, JES exceeds the time budget of 12 hours per run, which is the same time budget applied to all other methods. We will include these new results (plots across all iterations) in the revised version of the paper.
>
> > The greedy selection for the batch of points to evaluate is not the best option. Although the Hypervolume Subset Selection Problem is #P-hard, there are better alternatives (Guerreiro et al. (2021)).
>
> We would like to argue that the alternative Hypervolume Subset Selection Problem (HSSP) algorithms mentioned in Guerreiro et al. (2021) *may not be directly suitable* for our batch selection strategy. As the reviewer did not specify a particular HSSP method, we are unsure which algorithm should be the main focus in our response. To the best of our knowledge, these HSSP algorithms *do* *not guarantee satisfaction of a key criterion* in our design: the selected batch points must come from different exploration spaces $\mathcal T$ (Section 4.4, line 266). This criterion is essential for promoting diversity in the estimated Pareto front. Additionally, we find a practical concern regarding the *computational cost* of these HSSP algorithms. Specifically, as discussed in Section 6 of Guerreiro et al. (2021), for problems with more than 3 objectives, selecting the optimal subset generally requires evaluating all $N=\binom{b}{N_c}$ combinations of $b$ points among the $N_c$ candidate solutions. In our experimental setup, where $N_c=2000$, $N$ increases exponentially with the batch size (up to $b=10$), making HSSP computationally expensive for large batches. Nonetheless, we appreciate the reviewer’s suggestion and agree that, with further comprehensive investigation, HSSP methods may be adapted to align with the specific constraints of our approach.

---

### Official Review · Reviewer_51fE · 2025-06-30

**Clarity:** 4
**Significance:** 3
**Originality:** 3
**Rating:** 5
**Confidence:** 3

**Summary:**

This paper proposes a novel algorithm for multi-objective Bayesian optimization (MOBO) that also supports batched evaluations. MOBO is the problem setting in which one aims to optimize multiple, potentially competing objectives. Therefore, instead of searching for solutions that minimize all objectives, one aims to find Pareto optimal solutions, that is, solutions where one cannot find a better solution for an objective without worsening another objective.

Several algorithms for MOBO have been proposed, but they are often either not scalable in the number of objectives or suffer from high computational cost.

This paper aims to fill this gap by proposing an algorithm that is more scalable in the number of objectives and does not exhibit high computational cost. To this end, it approximates the convex hull of individual object minima (CHIM) to define search directions orthogonal to the CHIM. These orthogonal search directions are then used to find Pareto-optimal points, each of which is further augmented to find a dense set of points close to the Pareto front.

These points then serve as candidates for the next Bayesian optimization iteration. Batch BO is realized by conditioning the GP on previously selected points, using the posterior mean as a hallucinated outcome (Kriging believer).

The proposed method is evaluated on a wide range of benchmarks and shows promising performance.

**Questions:**

* How many objectives can MOBO-OSD handle? Can you construct (synthetic) benchmarks with arbitrarily many objectives?
* What's the significance of the batching? Can't the Kriging believer be combined with other MOBO AFs as well?

**Ethical Concerns:**

["NO or VERY MINOR ethics concerns only"]

**Final Justification:**

This paper provides an effective and well-designed solution for a relevant problem. I see room for improvement in the empirical evaluation and in the presentation of the batching mechanism but recommend the acceptance of the paper.

**Limitations:**

Yes

**Quality:**

3

**Strengths And Weaknesses:**

__Strengths__

* The paper is excellently written, and the presentation of the relatively complex algorithm is clear.
* The algorithm outperforms the competitors throughout the bank.
* The paper has high-quality figures that help understand the algorithm.

__Weaknesses__

* The methodology for batching seems disconnected from the paper and is orthogonal to the multi-objective design.
* Not all runs in the empirical evaluation have converged or are close to convergence, so it's not always clear if MOBO-OSD actually beats the other methods in the end.

__Minor Comments__

* Typo in the Figure in Section 4.1 ('Inidividual')
* The small Figures in 4.1 and 4.2 also deserve a number and a caption.

---

> ### Author Rebuttal · Authors · 2025-07-31
>
> We thank the reviewer for their insightful review. We address the questions and comments as follows:
>
> # Questions
>
> > How many objectives can MOBO-OSD handle? Can you construct (synthetic) benchmarks with arbitrarily many objectives?
>
> MOBO-OSD is designed to handle *any arbitrary number of objectives*, as it does not require any specific data structure for the objective space and is still maintained feasible computational cost even in batch settings. While it is possible to construct synthetic benchmark functions that scale to arbitrarily many objectives, such as the DTLZ series [16], we follow the standard practice in many MOBO papers [5, 22, 48, 12, 13, 31] by performing evaluation using benchmark problems with up to 6 objectives and batch size of 10. Note that several state-of-the-art methods face scalability limitations under these settings. For example, DGEMO [31] requires a carefully designed performance buffer tailored to each specific number of objectives, and its current open-access implementation cannot scale beyond 3 objectives. Similarly, qEHVI [12] becomes prohibitively expensive in batch settings, as demonstrated in the runtime analysis provided in Tables 4, 5, 6 and 7 of Appendix A.9.
>
> > What's the significance of the batching? Can't the Kriging believer be combined with other MOBO AFs as well?
>
> As mentioned in Section 1 (line 31), batching plays an important role in reducing the time and cost of expensive black-box optimization by *enabling parallel evaluations of objective functions*. Therefore, it is common practice for MOBO algorithms to be able to support batch settings [1, 5, 12, 31]. Regarding Kriging Believer (KB), it generally cannot be applied to other MOBO AFs because many MOBO AFs have their own batch mechanisms. For example, the qEHVI [12] AF sequentially integrates over unobserved data points when working with batch setting.
>
> # Weakness
>
> > The methodology for batching seems disconnected from the paper and is orthogonal to the multi-objective design.
>
> We would like to emphasize that our methodology for batching (batch selection strategy) is tightly integrated with the core design of our proposed method. Specifically, *the batching mechanism directly follows the construction of OSD subproblems*: each point in the batch is associated with a distinct OSD and its corresponding exploration space $\mathcal{T}$, as described in Section 4.4 and formalized in Eq. (4). This design ensures the batch maintains both strong hypervolume performance and diversity across the Pareto front. Therefore, the batching methodology is not orthogonal but rather fundamentally coupled to the multi-objective design.
>
> > Not all runs in the empirical evaluation have converged or are close to convergence, so it's not always clear if MOBO-OSD actually beats the other methods in the end.
>
> In MOBO, where objective functions are *expensive* and *black boxes*, the goal is to find the Pareto set of optimal solutions using the least number of function evaluations [29, 5]. Figures 2, 3, 8 and 9 show that MOBO-OSD consistently find better Pareto sets - measured by the hypervolume indicator - and uses fewer evaluation budget to reach such performance compared to all baselines. It is also worth noting that an evaluation budget of 200 iterations is standard practice in recent state-of-the-art MOBO works [5, 22, 48, 12, 31].
>
> ## Minor Comments
>
> Thank you for the comments. We will fix these in the revised version of the paper.

---

> > ### Comment · Reviewer_51fE · 2025-08-04
> >
> > I'd like to thank the authors for their reply.
> >
> > > MOBO-OSD is designed to handle any arbitrary number of objectives
> >
> > Do you expect empirical problems for a high number of objectives? If not, why not run a benchmark to show that MOBO-OSD actually can handle a high number of objectives?
> >
> > > Regarding Kriging Believer (KB), it generally cannot be applied to other MOBO AFs because many MOBO AFs have their own batch mechanisms.
> >
> > I understand that qEHVI and other MOBO AFs might come with their own batching mechanism, but that does not mean that they can't be combined with the Kriging Believer (KB). What hinders me from running qEHVI with $q=1$, adding a fantasized observation with the predicted mean for each objective to the model, and running qEHVI with $q=1$ again in order to obtain a second candidate for a batch?

---

> ### Author Response · Authors · 2025-08-06
>
> > Do you expect empirical problems for a high number of objectives? If not, why not run a benchmark to show that MOBO-OSD actually can handle a high number of objectives?
>
> Although MOBO-OSD can handle a high number of objectives, we expect significant computational cost ($M>10$). This issue is due to the HV computation, which is well-known to be computationally expensive in the many-objective setting - a limitation commonly discussed in prior literature of MOO [A, B, C]. This inherently affects the HV-based MOBO algorithms, including our proposed algorithm MOBO-OSD and other state-of-the-art MOBO algorithms such as qEHVI [12].
>
> As suggested by the reviewer, we conducted an experiment to run MOBO-OSD using the DTLZ2 benchmark problem with 10 and 20 objectives. The average computation cost for each iteration was approximately 280 and 1800 seconds for 10 and 20 objectives, respectively, of which the HV computation cost accounted for 250 and 1700 seconds. Note that, as shown in Table 7 of the manuscript, for smaller objective numbers $M=\\{2,\dots 4\\}$, the runtime is significantly lower, ranging from 1.13 to 2.73 seconds per iteration. We also attempted to run MOBO-OSD using the DTLZ2 benchmark problem with 50 objectives. While other components, such as GP training and solving OSD subproblems, took only approximately 200 seconds per iteration, the HV computation became infeasible: using pymoo [7], the process exceeded 5 hours without completion, and using botorch [3] resulted in an out-of-memory error on a machine with 128GB of RAM. It’s worth emphasizing again that to the best of our knowledge, MOBO algorithms typically consider problems with up to 6 objectives as in our paper [5, 31, 1, 12, 22, 48]. Nevertheless, scaling MOBO-OSD to high number of objectives remains an important challenge and is a promising direction for our future work.
>
> > I understand that qEHVI and other MOBO AFs might come with their own batching mechanism, but that does not mean that they can't be combined with the Kriging Believer (KB). What hinders me from running qEHVI with $q=1$, adding a fantasized observation with the predicted mean for each objective to the model, and running qEHVI with again in order to obtain a second candidate for a batch?
>
> Yes, it is practically possible to combine KB with qEHVI in the way mentioned by the reviewer. However, that would effectively remove the batching strategy proposed by the authors of qEVHI, which integrates over the uncertainty of unobserved data point in the batch [12]. In fact, qEHVI’s batching mechanism empirically outperforms several common batching mechanisms, one of which is related to KB (the “Posterior Mean” variant) [12]. This is why we argue KB is not generally applicable to other MOBO AFs, as KB would need to replace existing batching mechanisms of the MOBO AFs, which is not guaranteed to improve those AFs’ performance. On the other hand, our HVI acquisition function is a simple HV difference computation for the expected function values (the posterior mean), hence applying KB can benefit the performance by improving the predictive accuracy. Note that in our batching mechanism, apart from KB, we need to incorporate additional criterion w.r.t the exploration space to maintain diversity of the Pareto front.
>
> [A] Shang, Ke, and Hisao Ishibuchi. "A new hypervolume-based evolutionary algorithm for many-objective optimization." *IEEE Transactions on Evolutionary Computation* 24.5 (2020): 839-852.
>
> [B] Pang, Yong, et al. "An expensive many-objective optimization algorithm based on efficient expected hypervolume improvement." *IEEE Transactions on Evolutionary Computation* 27.6 (2022): 1822-1836.
>
> [C] Belakaria, Syrine, Aryan Deshwal, and Janardhan Rao Doppa. "Max-value entropy search for multi-objective Bayesian optimization." *Advances in neural information processing systems* 32 (2019).

---

### Official Review · Reviewer_6cNg · 2025-07-03

**Clarity:** 3
**Significance:** 2
**Originality:** 2
**Rating:** 5
**Confidence:** 3

**Summary:**

This paper introduces MOBO-OSD, a novel algorithm for batch multi-objective Bayesian optimization (MOBO). The core idea is to adapt the Normal Boundary Intersection (NBI) method to the limited-budget BO setting. The algorithm works by first defining an approximated convex hull of individual objective minima (CHIM). It then generates a set of well-distributed Orthogonal Search Directions (OSDs) and solves a series of constrained single-objective subproblems along these directions to identify candidate solutions on the Pareto front. To enhance solution density, the method incorporates a Pareto Front Estimation (PFE) technique to explore the local neighborhood of these candidates. Finally, a batch selection strategy using the Hypervolume Improvement acquisition function, Kriging Believer, and a diversity-promoting mechanism is employed to select points for parallel evaluation. Extensive experiments show that MOBO-OSD consistently outperforms state-of-the-art baselines on a variety of problems.

**Questions:**

1. Could you further elaborate on the fundamental advantages of the OSD-based subproblem formulation compared to a well-designed scalarization method that uses a similarly diverse set of weight vectors for batch selection? While empirically superior, the conceptual distinction could be clarified.

2. Your approach of searching along 1D Orthogonal Search Directions in the objective space is conceptually similar to methods that use 1D subspaces for high-dimensional single-objective BO (e.g., Kirschner et al., 2019; BOIDS). Have you considered this connection, and do you see any potential for cross-pollination of ideas between these domains?

**References**
- Ngo, Lam, Huong Ha, Jeffrey Chan, and Hongyu Zhang. "BOIDS: High-Dimensional Bayesian Optimization via Incumbent-Guided Direction Lines and Subspace Embeddings." In Proceedings of the AAAI Conference on Artificial Intelligence, vol. 39, no. 18, pp. 19659-19667. 2025.
- Kirschner, Johannes, Mojmir Mutny, Nicole Hiller, Rasmus Ischebeck, and Andreas Krause. "Adaptive and safe Bayesian optimization in high dimensions via one-dimensional subspaces." In International Conference on Machine Learning, pp. 3429-3438. PMLR, 2019.

**Ethical Concerns:**

["NO or VERY MINOR ethics concerns only"]

**Final Justification:**

This paper is recommended for acceptance. Its core contribution, MOBO-OSD, is a novel and effective algorithm for batch multi-objective Bayesian optimization. The method's strength lies in its concrete geometric approach: adapting Normal Boundary Intersection (NBI) with Orthogonal Search Directions (OSDs) to ensure a diverse and well-structured exploration of the Pareto front. This is intelligently combined with a Pareto Front Estimation (PFE) technique for density and a Kriging Believer strategy for efficient batch selection.

**Paper Formatting Concerns:**

No.

**Quality:**

2

**Strengths And Weaknesses:**

**Strengths**
- Novel Geometric Approach for Diversity: The paper's main strength is its novel adaptation of the NBI method to MOBO. By defining search directions geometrically (the OSDs), the algorithm has a structured and principled mechanism for generating a diverse set of candidate solutions, directly addressing a key challenge in MOBO.
- Effective Combination of Techniques: MOBO-OSD intelligently combines multiple powerful techniques. The use of Riesz s-Energy to generate well-distributed OSDs, the PFE technique to enrich the candidate pool without solving an excessive number of subproblems, and a thoughtful batch selection strategy all contribute to a robust and effective algorithm.
- Strong and Comprehensive Empirical Validation: The authors conduct extensive experiments on nine benchmark problems (with two to six objectives) against seven state-of-the-art baselines in both sequential and batch settings. The results consistently demonstrate the superiority of MOBO-OSD in terms of hypervolume difference, which is a strong piece of evidence for the method's effectiveness.
- Designed for Batch Optimization: The algorithm is designed from the ground up to support batch optimization. The batch selection strategy, which combines Kriging Believer with a mechanism to ensure candidates are drawn from different exploration zones, is a practical and well-reasoned approach to accelerate the optimization process with parallel resources.

**Weakness**
-  Incomplete Contextualization within the MOBO Landscape: The paper's positioning could be improved with a broader discussion of the existing MOBO literature.

      - Conceptual Overlap with Scalarization: The core mechanism involves solving multiple single-objective subproblems. While the generation of these subproblems is geometrically motivated by OSDs, the paper could provide a deeper discussion on how this fundamentally differs from advanced scalarization techniques that also generate diverse weight vectors for batch optimization.
      -  Omission of related MOBO baseline: The literature review focuses entirely on methods that approximate the full Pareto front but omits the following work: Zuluaga, Marcela, Andreas Krause, and Markus Püschel. "e-pal: An active learning approach to the multi-objective optimization problem." Journal of Machine Learning Research 17, no. 104 (2016): 1-32.

- Limited Discussion of Related Single-Objective Methods: The paper misses an opportunity to connect its directional search approach to similar ideas in high-dimensional single-objective BO. Methods like those by Kirschner et al. (2019) and Ngo et al. (2025) also leverage 1D line searches to make complex problems tractable. Discussing these parallels would better situate MOBO-OSD within the broader BO literature.
- Algorithmic Complexity and Hyperparameters: The complete MOBO-OSD algorithm is a multi-stage pipeline with several components and associated hyperparameters . While an ablation study on $n_\beta$ is provided, a more comprehensive analysis of the method's complexity and sensitivity to these different moving parts would be beneficial.

---

> ### Author Rebuttal · Authors · 2025-07-31
>
> We thank the reviewer for their insightful review. We address the questions and comments as follows:
>
> # Questions
>
> > 1. Could you further elaborate on the fundamental advantages of the OSD-based subproblem formulation compared to a well-designed scalarization method that uses a similarly diverse set of weight vectors for batch selection? While empirically superior, the conceptual distinction could be clarified.
>
> There are several advantages of our OSD subproblem formulation compared to the most closely related scalarization technique - linear scalarization (LS) [37]. First, *LS generates search directions at random*, which can be less efficient than the data-driven search directions proposed in our OSD formulation. Second, even when LS weight vectors are well-distributed (e.g., using Riesz s-Energy [8]), *LS is limited to finding solutions on the convex regions of the Pareto front* [A]. In contrast, our proposed OSD formulation is capable of identifying solutions on the Pareto front of arbitrary shape, including both convex and non-convex regions. Note that due to this limitation, LS has been superseded by the most widely used scalarization technique - Tchebychev scalarization (TS) [37]. TS can handle non-convex PF and has been widely selected in other works when evaluating scalarization techniques [29, 37, 12, 31]. However, one critical drawback of TS lies in its non-smooth formulation caused by the maximization operator, which makes TS suffers from non-differentiability and slow convergence [B]. On the other hand, the MOBO-OSD subproblem formulation is differentiable, either analytically under common kernels or by automatic differentiation via computational graphs, as described in Appendix A.4. Empirically, our proposed method consistently outperforms baselines employing TS, such as qParEGO. Figures 2, 3, 8 and 9 show that MOBO-OSD outperforms qParEGO across different benchmark functions with diverse PF characteristics, including convex (ZDT1) and concave (DTLZ2). We will include this discussion in the revised version of the paper.
>
> > 2. Your approach of searching along 1D Orthogonal Search Directions in the objective space is conceptually similar to methods that use 1D subspaces for high-dimensional single-objective BO (e.g., Kirschner et al., 2019; BOIDS). Have you considered this connection, and do you see any potential for cross-pollination of ideas between these domains?
>
> LineBO (Kirschner et al. (2019)) and BOIDS (Ngo et al. (2025)) are fundamentally different approaches and are not directly compatible with our method. The primary distinction lies in the formulation of these 1D lines: both LineBO and BOIDS construct search directions with some degree of *randomness*, whereas MOBO-OSD relies on *deterministic* OSD directions that are orthogonal to the approximated CHIM. Additionally, another key difference is in the search domain: LineBO and BOIDS define 1D lines in the *input space*, while our MOBO-OSD formulates 1D search directions in the *output space*. Furthermore, as noted by the reviewer, LineBO and BOIDS are primarily designed to *make high-dimensional problems tractable*, while MOBO-OSD focuses on the *generating a well-distributed set of Pareto optimal points* on the Pareto front. For these reasons, we argue that cross-pollinating OSD with uniformly random directions (as in LineBO) or incumbent-guided directions (as in BOIDS) might not be feasible in the MOBO problems, as it would compromise the core purpose of OSD: producing orthogonal directions to the approximated CHIM to ensure well-distributed coverage of the Pareto front.
>
> # Weakness
>
> > Incomplete Contextualization within the MOBO Landscape: The paper's positioning could be improved with a broader discussion of the existing MOBO literature…
>
> Regarding the conceptual overlap with scalarization, please refer to our response to Question 1 for a discussion of scalarization techniques. We will include this discussion in the revised version of the paper.  Regarding the omission of the related MOBO baseline ($\\varepsilon$-PAL), we would like to emphasize that $\\varepsilon$-PAL is specifically designed for *finite discrete input spaces*, which is why it was not mentioned. Furthermore, prior studies [37, 5, 22] have highlighted that $\\varepsilon$-PAL shares methodological similarities with [C], which has been empirically outperformed by our baseline USeMO [5]. Nevertheless, we will include a brief discussion of $\\varepsilon$-PAL in the revised version of the paper.
>
> > Limited Discussion of Related Single-Objective Methods: The paper misses an opportunity to connect its directional search approach to similar ideas in high-dimensional single-objective BO. Methods like those by Kirschner et al. (2019) and Ngo et al. (2025) also leverage 1D line searches to make complex problems tractable. Discussing these parallels would better situate MOBO-OSD within the broader BO literature.
>
> Please see the response to Question 2 for detailed explanation. We will include this discussion to the revised version of the paper.
>
> > Algorithmic Complexity and Hyperparameters: The complete MOBO-OSD algorithm is a multi-stage pipeline with several components and associated hyperparameters. While an ablation study on $n_\beta$ is provided, a more comprehensive analysis of the method's complexity and sensitivity to these different moving parts would be beneficial.
>
> Regarding the algorithmic complexity, we would like to note that a comprehensive runtime comparison between MOBO-OSD and the baselines is already provided in Tables 4, 5, 6 and 7 in Appendix A.9, covering all benchmark problems and batch settings $b=\\{1,4,8,10\\}$. In addition, we have now conducted a theoretical time complexity analysis, summarized as follows. The time complexity is $\\mathcal O(NM)$ for the approximated CHIM and OSD formulation, $\\mathcal O(D^3+Nn+N^2n)$ for solving each MOBO-OSD subproblem, and  $\\mathcal O(D^3+Nn)$ for the Pareto Front Estimation step, where $N$ and $n$ denote the number of surrogate model training and testing points, respectively, $D$ and $M$ denote the number of dimensions and objectives, respectively. We will include this theoretical time complexity analysis in the revised version of the paper.
>
> Regarding the sensitivity analysis of the hyperparameters in MOBO-OSD’s components, we presented two ablation studies for the two most important components of MOBO-OSD in Section 5.1.2. The first ablation study targets the OSD component, specifically examining the impact of the number of OSDs, $n_\\beta$, on overall performance. Figures 4 and 6 demonstrate that the number of OSDs, $n_\\beta$, has minimal effect on MOBO-OSD’s overall performance, indicating the robustness of the method. The second ablation study focuses on the Pareto Front Estimation (PFE) component, evaluating its necessity within MOBO-OSD. Table 1 demonstrates that although PFE enhances the efficiency of MOBO-OSD, it is not a required component, as comparable performance can still be achieved by increasing the density of the $\\mathcal{U}(\\boldsymbol\\beta)$ set. We will consider adding more sensitivity analysis for other minor components in the revised version of the paper.
>
> ## References
>
> [A] Zhang, Richard, and Daniel Golovin. "Random hypervolume scalarizations for provable multi-objective black box optimization." *International conference on machine learning*. PMLR, 2020.
>
> [B] Lin, Xi, et al. "Smooth Tchebycheff Scalarization for Multi-Objective Optimization." *International Conference on Machine Learning*. PMLR, 2024.
>
> [C] Ponweiser, Wolfgang, et al. "Multiobjective optimization on a limited budget of evaluations using model-assisted-metric selection." *International conference on parallel problem solving from nature*. Berlin, Heidelberg: Springer Berlin Heidelberg, 2008.

---

> > ### Comment · Reviewer_6cNg · 2025-08-07
> >
> > I appreciate the author's detailed response. One thing I want to clarify about question 2 is that I was looking for some potential link, for example, in the high-dimensional multi-objective tasks, where the input space and output space 1D search could form an end-to-end 1D solution. I believe the answer addresses my initial concerns. I'll increase my score accordingly.

---

> > > ### Author Response · Authors · 2025-08-07
> > >
> > > We would like to thank the reviewer for acknowledging our response, and we are glad that our clarification addressed your concerns.

---

### Note · Authors · 2025-08-13

We would like to thank all the reviewers for their insightful and helpful comments. We received four constructive and positive reviews, and we responded to each review in detail. Three reviewers expressed satisfaction with our answers, and one reviewer requested further clarification, which we also provided.

During the rebuttal period, we provided various discussions, such as the relationship between our proposed method MOBO-OSD and other scalarization techniques (Reviewer 6cNg), the practical challenges of MOBO-OSD in settings with a large number of objectives (Reviewer 51fE), and the limitations in noisy settings (Reviewer RbX4). As suggested by Reviewer 6cNg’s, we added a *theoretical time complexity analysis* for each MOBO-OSD component. Furthermore, following Reviewer xpJo’s recommendations, we conducted additional experiments using *three new metrics* (IGD, IGD+, $\varepsilon$-indicator) and *one new baseline* (JES), with results showing that MOBO-OSD still outperforms under these extended evaluations.

We believe our responses address all reviewers’ concerns, and we will incorporate them into the revised version of the paper.

---

### Decision · Program_Chairs · 2025-09-17

**Decision:**

Accept (poster)

**Comment:**

In this paper, the authors propose NBI-MOBO, a multi-objective Bayesian Optimization algorithm inspired by the Normal Boundary Intersection framework. NBI-MOBO decomposes the multi-objective problem into a series of single-objective subproblems, referred to as NBI-MOBO subproblems, each constrained along a predefined search direction in the objective space. The experiments and analysis on various synthetic and real-world benchmark functions with various objective functions, ranging from 2 to 6 objectives, show that NBI-MOBO can outperform sota methods. According to the reviewers, the paper's main strength is its novel adaptation of the NBI method to the MOBO setting. The paper has a strong and comprehensive empirical validation. Moreover, the paper is excellently written. Furthermore, the proposed method outperforms all other methods on all the benchmarks considered. Finally, the work is clearly novel. Thus, I belive this is a nice and interesting paper that will get the attention of the BO community.